# Cell cycle arrest induces lipid droplet formation and confers ferroptosis resistance

Hyemin Lee [1], Amber Horbath [1], Lavanya Kondiparthi[2,7], Jitendra Kumar Meena [3], Guang Lei [1], Shayani Dasgupta[1], Xiaoguang Liu [1], Li Zhuang[1], Pranavi Koppula [1,4], Mi Li [1], Iqbal Mahmud [5], Bo Wei[5], Philip L. Lorenzi [5], Khandan Keyomarsi [1,4], Masha V. Poyurovsky[2,8], Kellen Olszewski[2,9] & Boyi Gan [1,4,6] ✉

How cells coordinate cell cycling with cell survival and death remains incompletely understood. Here, we show that cell cycle arrest has a potent suppressive effect on ferroptosis, a form of regulated cell death induced by overwhelming lipid peroxidation at cellular membranes. Mechanistically, cell cycle arrest induces diacylglycerol acyltransferase (DGAT)–dependent lipid droplet formation to sequester excessive polyunsaturated fatty acids (PUFAs) that accumulate in arrested cells in triacylglycerols (TAGs), resulting in ferroptosis suppression. Consequently, DGAT inhibition orchestrates a reshuffling of PUFAs from TAGs to phospholipids and re-sensitizes arrested cells to ferroptosis. We show that some slow-cycling antimitotic drug–resistant cancer cells, such as 5-fluorouracil–resistant cells, have accumulation of lipid droplets and that combined treatment with ferroptosis inducers and DGAT inhibitors effectively suppresses the growth of 5-fluorouracil–resistant tumors by inducing ferroptosis. Together, these results reveal a role for cell cycle arrest in driving ferroptosis resistance and suggest a ferroptosis-inducing therapeutic strategy to target slow-cycling therapy-resistant cancers.

Fatty acids are essential building blocks of cellular membranes as well as important fuels for energy metabolism. Cells can obtain fatty acids via de novo synthesis, uptake from circulation (or medium), and/or breakdown of existing phospholipids (PLs). When nutrients are abundant, cells store excess fatty acids in the form of neutral lipids (such as triacylglycerols [TAGs]) in lipid droplets, organelles that are surrounded by a PL monolayer and play an important role in lipid storage[1,2]. Diacylglycerol acyltransferases (DGATs; including DGAT1 and DGAT2) catalyze the last and committed step in TAG synthesis by adding a fatty acyl group to the *sn-3* position of diacylglycerol to form

TAG[3,4]. Fatty acids stored in lipid droplets later can be used for cellular membrane construction and energy metabolism when cells encounter harsh cellular environments (such as cell starvation). Lipid droplets have also been shown to protect cells from lipotoxicity by sequestering TAGs containing polyunsaturated fatty acids (PUFAs, which contain more than one double bond)[5,6]. Therefore, in this context, lipid droplets play a cytoprotective role.

PUFA-containing PLs, such as arachidonic acid–containing and adrenic acid–containing phosphatidylethanolamines (PEs), are particularly susceptible to peroxidation and render cells sensitive to

[1]Department of Experimental Radiation Oncology, The University of Texas MD Anderson Cancer Center, Houston, TX 77030, USA. [2]Kadmon Corporation, New York, NY 10016, USA. [3]Verna and Marrs McLean Department of Biochemistry and Molecular Biology, Baylor College of Medicine, Houston, TX 77030, USA. [4]The University of Texas MD Anderson Cancer Center UTHealth Houston Graduate School of Biomedical Sciences, Houston, TX 77030, USA. [5]Metabolomics Core Facility, Department of Bioinformatics and Computational Biology, The University of Texas MD Anderson Cancer Center, Houston, TX 77030, USA. [6]Department of Molecular and Cellular Oncology, The University of Texas MD Anderson Cancer Center, Houston, TX 77030, USA. [7]Present address: Sanofi US, Cambridge, MA 02139, USA. [8]Present address: PMV Pharmaceuticals, Princeton, NJ 08540, USA. [9]Present address: Carl Icahn Labs, Princeton University, Princeton, NJ 08544, USA. ✉e-mail: bgan@mdanderson.org

ferroptosis, a type of regulated cell death that is triggered by uncontrolled PL peroxidation[7,8]. Iron-dependent peroxidation of membrane PL damages membrane integrity and, if left unrepaired, eventually causes ferroptosis[9–11]. For example, treatment with ferroptosis inducers, such as erastin and RSL3, significantly compromises cellular defense systems against ferroptosis and unleashes potent ferroptosis in many cancer cell lines[8,12]. In contrast to PUFAs, incorporation of monounsaturated fatty acids (fatty acids containing one double bond) into PLs suppresses lipid peroxidation and ferroptosis, presumably by displacing PUFAs from PLs on cellular membranes[13]. In addition, previous studies demonstrated that supplementation of exogenous fatty acids and inhibition of fatty acid synthesis alter lipid metabolism and affect ferroptosis sensitivity[14,15]. Therefore, the cellular lipid composition plays a critical role in regulating ferroptosis.

Whereas the role of lipid metabolism in regulating ferroptosis is well established, whether and, if so, how different cell states affect lipid composition and ferroptosis sensitivity remain much less understood[16]. In this study, we uncovered a previously unappreciated suppressive effect of cell cycle arrest on ferroptosis, and showed that it suppresses ferroptosis at least partly by promoting DGAT-dependent lipid droplet formation to sequester excessive PUFAs in TAGs. We further revealed that slow-cycling chemoresistant and radioresistant cancer cells exhibit elevated levels of lipid droplets and increased ferroptosis resistance and that combined treatment with ferroptosis inducers and DGAT inhibitors overcomes ferroptosis resistance and suppresses the growth of such therapy-resistant tumors.

## Results

### Cell cycle arrest drives ferroptosis resistance

To determine whether cell cycle arrest has any impact on ferroptosis sensitivity, we treated Caki-1 cells (a human renal cell carcinoma cell line) with different cell cycle inhibitors, including hydroxyurea (to arrest cells at early S phase), thymidine (to arrest cells at the G1/S boundary), and colcemid and nocodazole (to arrest cells at G2/M phase) (Fig. 1a). Notably, we found that arresting cells with any of these cell cycle inhibitors potently suppressed ferroptosis induced by treatment with erastin or RSL3 (Fig. 1b, c and Supplementary Fig. 1a). Lipid peroxidation is a hallmark of ferroptosis[10,11]. Correspondingly, erastin- or RSL3-induced lipid peroxidation was suppressed under these cell cycle–arrested conditions (Supplementary Fig. 1b, c). We confirmed these findings using other renal cell carcinoma cell lines ACHN, 786-O, and TK10 (Fig. 1d and Supplementary Fig. 1d–l), the fibrosarcoma cell line HT-1080, the melanoma cell line A375, and immortalized mouse embryonic fibroblasts (MEFs) (Fig. 1e–g and Supplementary Fig. 1m–o), thereby establishing the generalizability of our findings across different cellular contexts. Of note, treatment with cell cycle inhibitors also triggered obvious cell death in some cell lines. However, this cell death was not ferroptosis, as treatment with the ferroptosis inhibitor ferrostatin-1 failed to block it (for example, see Fig. 1f, g). We further showed that, after cell cycle inhibitors were removed from the medium, cells regained sensitivity to erastin-induced ferroptosis (Fig. 1h), suggesting ferroptosis suppression by cell cycle arrest is reversible.

To substantiate our findings described above, we tested the effect of treatment with palbociclib, a cyclin-dependent kinase (CDK) 4/6 inhibitor, on ferroptosis sensitivity. We found that palbociclib arrested cells at G0/G1 phase (Fig. 1i) as expected and protected cells from erastin- or RSL3-induced lipid peroxidation and ferroptosis (Fig. 1j, k and Supplementary Fig. 1p, q). Because CDK4/6-mediated phosphorylation of retinoblastoma (Rb) protein is a critical step in S-phase entry[17], we further sought to determine whether the effect of palbociclib on ferroptosis suppression depends on Rb. To this end, we generated Rb-deficient cells using the CRISPR-Cas9 approach (Fig. 1l). Rb deficiency largely abrogated palbociclib-induced cell cycle arrest (Supplementary Fig. 1r) and ferroptosis suppression (Fig. 1m),

suggesting that an intact Rb checkpoint is required for ferroptosis suppression induced by CDK4/6 inhibition. This result further established that the effect of CDK4/6 inhibition on ferroptosis suppression is mediated through its function in inducing cell cycle arrest.

Our data on cell cycle inhibitors suggested that the effect of cell cycle arrest on ferroptosis is independent of the arrested cell cycle phases (Fig. 1a–c). To further examine this, we generated CDK1-deficient cells (Fig. 1n). Similar to treatment with colcemid or nocodazole, CDK1 deficiency arrested cells at the G2/M phase (Fig. 1o) and rendered cells remarkably resistant to erastin- or RSL3-induced lipid peroxidation and ferroptosis (Fig. 1p, q and Supplementary Fig. 1s, t). We therefore concluded that cell cycle arrest suppresses ferroptosis and that this effect occurs irrespectively of the phase at which cells are arrested.

### Cell cycle arrest increases TAG levels and induces lipid droplet accumulation

To investigate the underlying mechanisms of cell cycle arrest in ferroptosis resistance, we examined whether cell cycle arrest affects several known ferroptosis mechanisms, including 1) expression of the key ferroptosis-regulatory proteins ACSL4, GPX4, SLC7A11, FSP1, and DHODH (Supplementary Fig. 2a); 2) cystine uptake levels (Supplementary Fig. 2b); 3) intracellular glutathione levels (Supplementary Fig. 2c); and 4) labile iron levels (Supplementary Fig. 2d). We found that treatment with cell cycle inhibitors either did not affect the examined ferroptosis mechanisms or did but the change was not consistent with the ferroptosis resistance phenotype; for example, treatment with nocodazole decreased cystine uptake and glutathione levels (Supplementary Fig. 2b, c). This effect of nocodazole would be expected to promote ferroptosis, but cannot explain the ferroptosis suppression phenotype caused by nocodazole treatment (Fig. 1b, c).

Considering the central role of lipid metabolism in regulating ferroptosis, we next studied lipid compositional changes in cell cycle–arrested cells by conducting untargeted lipidomic analyses of Caki-1 cells treated with a vehicle or nocodazole. Treatment with nocodazole resulted in a global increase in the relative abundance of many lipid species, with the increase in TAGs and ether-TAGs (TAG-Os) being most significant (Fig. 2a, b). Cells convert excess lipids into neutral lipids, including TAGs and TAG-Os, and preserve them in lipid droplets[1,2]. Because cell cycle–arrested cells had most notable increases in these lipid species (Fig. 2a), we examined whether such cells also had increased accumulation of lipid droplets. Staining with BODIPY 493/503, a lipophilic probe that labels cellular neutral lipids localized on lipid droplets, revealed markedly increased staining of lipid droplets in cells treated with cell cycle inhibitors (Fig. 2c, d and Supplementary Fig. 3a, b). Lipid droplet levels in arrested cells decreased in a time-dependent manner upon release from cell cycle inhibitors, eventually reaching levels comparable to those in vehicle-treated control cells (Supplementary Fig. 3c, d), which is consistent with our earlier observation showing the reversibility of ferroptosis sensitivity in cells released from cell cycle inhibitors (Fig. 1h). Furthermore, treatment with the CDK4/6 inhibitor palbociclib increased lipid droplet levels in these cells (Fig. 2e), whereas Rb deficiency abolished CDK4/6 inhibition–induced lipid droplet formation (Supplementary Fig. 3e). Likewise, genetic ablation of CDK1 resulted in accumulation of lipid droplets (Fig. 2f). Further analyses confirmed a consistent elevation in TAG levels across these different cell cycle inhibition conditions (Fig. 2g).

Of note, the expression of both DGAT1 and DGAT2 was increased in cell cycle–arrested cells (Supplementary Fig. 3f, g). (In our studies, we showed that cell cycle inhibition increased DGAT2 mRNA levels. Our efforts to detect endogenous DGAT2 protein levels, in line with findings from other studies[18], proved unsuccessful.) Consistent with the central role of DGATs in TAG synthesis[3,4], lipidomic analyses revealed that treatment with T863 and PF06427878 (which inhibit

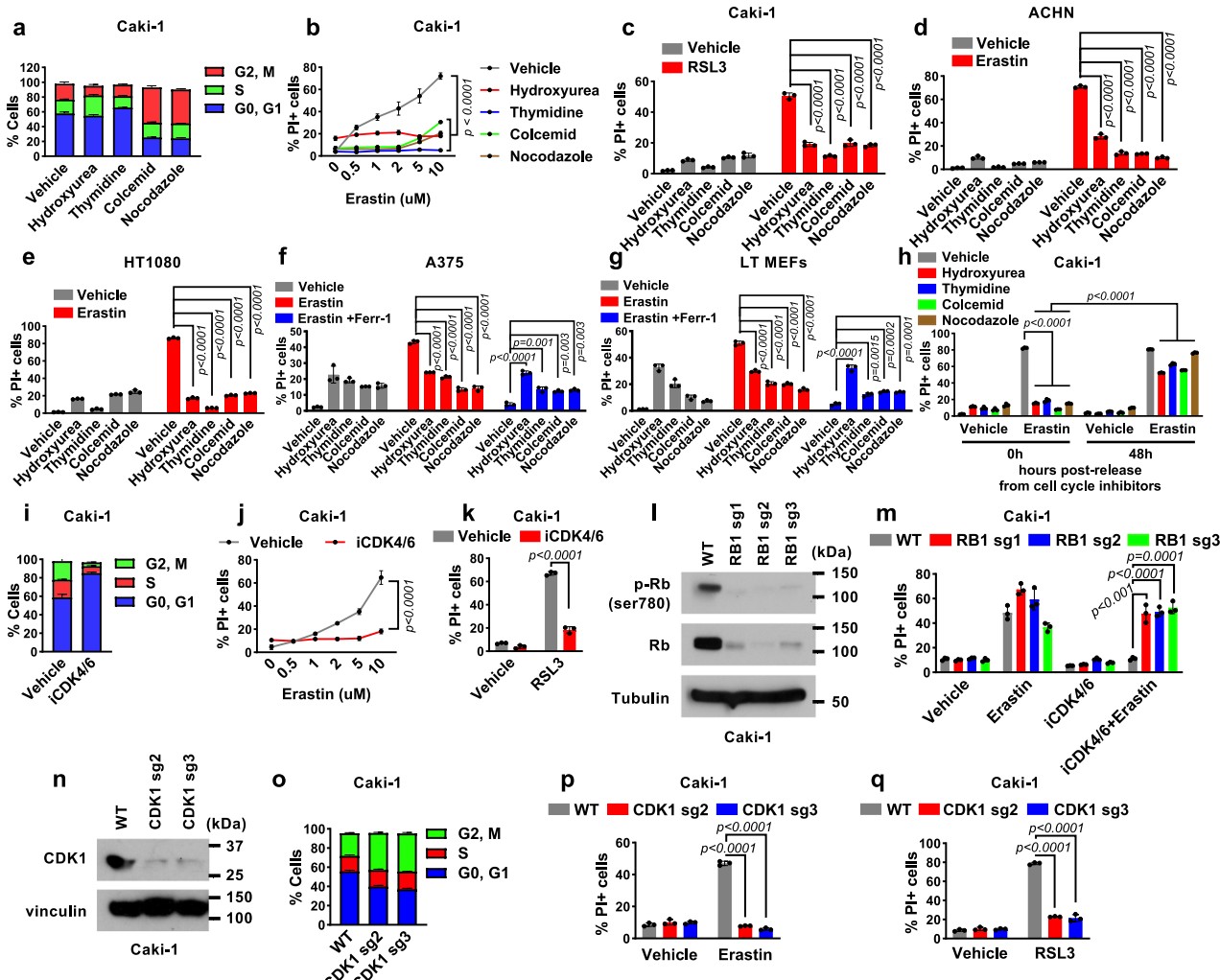

**Fig. 1 | Cell cycle arrest drives resistance to ferroptosis. a** Cell cycle analysis of Caki-1 cells treated with hydroxyurea (0.3 mM), thymidine (1 mM), colcemid (0.035 µg/ml), or nocodazole (200 nM) for 48 h. Quantification of PI-positive dead Caki-1 cells using flow cytometry after 24 h of pretreatment with cell cycle inhibitors followed by treatment with 0-10 µM erastin for 18 h (**b**) or 50 nM RSL3 for 24 h (**c**). PI-positive cell populations in cells pretreated with cell cycle inhibitors for 24 h followed by treatment with 5 µM erastin for 24 h in ACHN (**d**), 10 µM erastin for 24 h in HT1080 (**e**), 20 µM erastin for 24 h in A375 (**f**), and 2 µM erastin for 16 h in MEFs (**g**). **h** PI-positive cell population in cells treated with 2 µM erastin for 18 h. Cells were released from either vehicle, hydroxyurea or nocodazole for indicated time prior to erastin treatment. **i** Cell cycle analysis of Caki-1 cells treated with 2 µM iCDK4/6 for 48 h. PI-positive cells after 24 h of pretreatment with the iCDK4/6

followed by treatment with 0-10 µM erastin for 18 h (**j**) or 50 nM RSL3 for 24 h (**k**). **l** Immunoblot showing the levels of Rb Ser780 phosphorylation and Rb expression in cells with the indicated genotypes. **m** Populations of PI-positive Caki-1 cells with the indicated genotypes and treatment for 18 h. **n** Immunoblot of CDK1 in wild-type (WT) and CDK1 knockout Caki-1 cells. **o** Cell cycle analysis of WT and CDK1 knockout Caki-1 cells. PI-positive Caki-1 cells treated with 2 µM erastin for 18 h (**p**) or 50 nM RSL3 for 24 h (**q**). Mean ( ± SD) values are shown. *n* = 3. *n* indicates independent repeats, *P* values were calculated using two-way ANOVA (**b**, **j**) or an unpaired, two-tailed *t*-test. Different doses of cell cycle inhibitors were used with each cell line. Details on the drug treatment concentrations and times are provided in Supplementary Table 1. Source data are provided as a Source Data file.

DGAT1 and DGAT2, respectively; collectively referred to as iDGAT1/2 herein) led to anticipated outcomes in both vehicle- and nocodazole-treated cells: a decrease in TAG and TAG-O levels (consequently, iDGAT1/2 treatment normalized the increases in TAG and TAG-O levels caused by nocodazole), and a concomitant increase in dia-cylglycerol levels (Fig. 2h–l). In vehicle-treated cells (i.e., normal proliferating cells without nocodazole treatment), the impact of DGAT inhibition on other lipid species exhibited a dynamic nature, leading to increases in certain lipids while causing decreases in others (Fig. 2i). Notably, the impact of DGAT inhibition on the lipidome changed significantly under nocodazole treatment conditions; in this context, the primary lipid species that exhibited a decrease were TAG and TAG-O, whereas the levels of most other lipid species, including various PLs, displayed increases (Fig. 2j). A more detailed analysis focusing on PLs revealed a substantial increase in the levels of

multiple PUFA-PLs and PUFA-ether PLs (particularly in the form of PEs) following iDGAT1/2 treatment (Fig. 2k, l).

Consistently, we found that DGAT1/2 inhibition suppressed lipid droplet accumulation caused by cell cycle inhibition (Fig. 2m, n), with a more potent effect by DGAT1/2 co-inhibition than by either single inhibition (Supplementary Fig. 3h), suggesting that both DGAT1 and DGAT2 regulate lipid droplet formation under cell cycle inhibition conditions. Finally, we showed that genetic deletion of *DGAT1* and *DGAT2* in Caki-1 cells significantly reduced lipid droplet levels under hydroxyurea treatment (Fig. 2o, p). Taken together, these data suggest that cell cycle arrest increases TAG and lipid droplet levels in a DGAT-dependent manner, and that under nocodazole treatment conditions, DGAT inhibition does not seem to significantly affect de novo lipid synthesis but rather orchestrates a reshuffling of fatty acids, including PUFAs, from TAGs to various PLs.

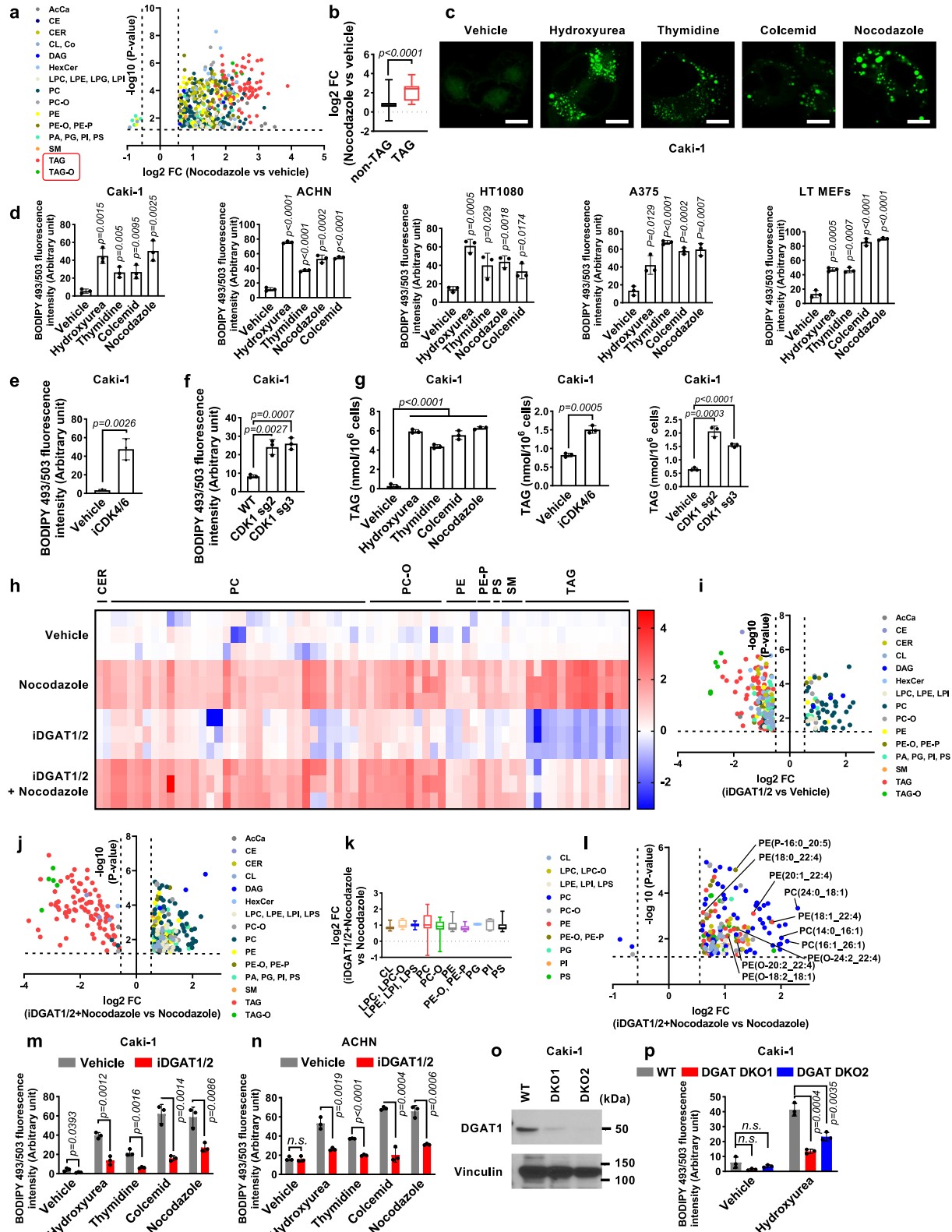

## DGATs mediate ferroptosis resistance caused by cell cycle inhibition

Based on our observations that cell cycle–arrested cells have increased levels of TAGs and lipid droplets as well as enhanced resistance to ferroptosis and that DGAT inhibition in cell cycle–arrested cells decreased TAG levels but substantially increased the levels of PUFA-PLs and PUFA-ether PLs (which are known substrates for lipid peroxidation[7,15]), we

hypothesized that cell cycle inhibition promotes ferroptosis resistance by enhancing DGAT-mediated TAG synthesis and lipid droplet formation; this process appears to entail the sequestration of PUFAs, preventing their incorporation into PLs and ether PLs. To test this hypothesis, we examined whether DGAT inactivation would reverse ferroptosis resistance phenotypes induced by cell cycle arrest. Treatment with iDGAT1/2 at least partly restored erastin-induced ferroptosis

**Fig. 2 | Cell cycle arrest induces lipid droplet accumulation. a** Volcano plot displays log2 fold-change (FC) in lipid species abundance, comparing nocodazole-treated to vehicle-treated Caki-1 cells ($n = 510$, $P < 0.05$, FC > 1.5). AcCa, acyl carnitine; CE, cholesterol ester; CER, ceramide; CL, cardiolipin; Co, coenzyme; DAG, diacylglyceride; HexCer, hexosyl ceramides; LPC, lysophosphatidylcholine; LPE, lysophosphatidylethanolamine; LPG, lysophosphatidylglycerol; LPI, lysophosphatidylinositol; PC, phosphatidylcholine; PC-O, ether-linked phosphatidylcholine; PE, phosphatidylethanolamine; PE-O/PE-P, ether-linked phosphatidylethanolamine; PA, phosphatidic acid; PG, phosphatidylglycerol; PI, phosphatidylinositol; PS, phosphatidylserine; SM, sphingomyelin; TAG, triacylglyceride; TAG-O, ether-linked triacylglyceride. **b** Boxplots displaying log2 FC in non-TAG ($n = 413$) and TAG ($n = 97$) lipid species abundance between nocodazole-treated and vehicle-treated Caki-1 cells, with medians and whiskers indicating min-max values. BODIPY 493/503 staining in Caki-1 cells after 48 h of cell cycle inhibitor treatment. scale bar, 20 μm (**c**), and quantifications in the indicated cell lines (**d**). BODIPY 493/503 staining in Caki-1 cells treated with iCDK4/6 for 48 h (**e**), or WT and sgCDK1 cells (**f**). **g** Triglyceride concentrations in Caki-1 cells. **h** Heat map of the significantly changed lipids (unpaired, two-tailed $t$-test; FDR-corrected $P < 0.05$, fold change > 1.5 in vehicle- versus nocodazole-treated Caki-1 cells). Rows represent samples, columns represent normalized intensities, color-coded from red (high intensity) to blue (low intensity). Volcano plots for iDGAT1/2-treated versus vehicle-treated (**i**, $n = 276$) and iDGAT1/2 and nocodazole-treated versus nocodazole-treated (**j**, $n = 302$) Caki-1 cells. Log2 FC in phospholipid abundance comparing iDGAT and nocodazole-treated to nocodazole-treated Caki-1 cells, depicted with boxplots (**k**) and volcano plots (**l**). $n = 192$. LPC-O, ether-linked lysophosphatidylcholine; LPS, lysophosphatidylserine. BODIPY 493/503 staining in Caki-1 (**m**) and ACHN (**n**) cells. **o** Immunoblot of DGAT1 in Caki-1 cells. **p** BODIPY 493/503 staining in WT, DGAT DKO1, and DKO2 Caki-1 cells with or without exposure to hydroxyurea (0.3 mM) for 48 h. Mean ($\pm$ SD) values are shown. $n = 3$. $n$ indicates independent repeats, except **a**, **b**, **i**, **j**, **k**, and **l**. unpaired, two-tailed $t$-test. n.s. not significant. Source data are provided as a Source Data file.

and lipid peroxidation under cell cycle inhibitor treatment conditions (Fig. 3a–h and Supplementary Fig. 4a–i). Consistent with their effect on lipid droplet formation (Supplementary Fig. 3h), co-treatment with DGAT1 and DGAT2 inhibitors (i.e., iDGAT1/2) had a greater ferroptosis restoration effect than did treatment with either inhibitor alone (Supplementary Fig. 4a). We made similar observations in cells with genetic deletion of *DGAT1/2* (Fig. 3i–p). Of note, our data demonstrated that pharmacological inhibition or genetic ablation of DGATs did not affect ferroptosis sensitivity in normal proliferating cells (i.e., cells not treated with any cell cycle inhibitor). This is consistent with our lipidomic analyses which revealed that DGAT inhibition resulted in PUFA-PL level increases specifically in nocodazole-treated cells, but not in normal proliferating cells (see Fig. 2i, j). We further examined the effect of iDGAT1/2 in palbociclib-treated or *CDK1*-deficient cells and confirmed that they resensitized the cells to ferroptosis (Fig. 3q–t).

Cells obtain fatty acids mainly via de novo synthesis and/or uptake from the extracellular environment (such as the medium for cultured cells). We found that when cells were cultured in lipoprotein-free fetal bovine serum, the accumulation of lipid droplets, the increase in DGAT expression, and the development of resistance to ferroptosis induced by cell cycle arrest were largely abolished (Supplementary Fig. 5a–f). Taken together, these data strongly suggest that cell cycle arrest suppresses ferroptosis at least partly through promotion of DGAT-mediated TAG synthesis and lipid droplet formation, and that the observed phenotypes related to DGAT, lipid droplet formation, and ferroptosis resistance in cell cycle–arrested cells primarily depend on the uptake of lipids from the extracellular environment.

## Slow-cycling therapy-resistant cancer cells have accumulation of lipid droplets

One of the major challenges in cancer therapy is a lack of effective strategies to eliminate residual therapy-resistant cancer cells, which survive after treatment with antimitotic drugs and eventually drive tumor recurrence[19,20]. Recent studies revealed that such therapy-resistant cells are often slow-cycling and therefore less susceptible to antiproliferative drugs than bulk tumor cells[21,22]. Our results described previously prompted us to examine whether slow-cycling therapy-resistant cells are also protected from ferroptosis owing to their high lipid droplet levels. Previous studies showed that 5-fluorouracil (5-FU) sensitivity and lipid droplet levels were inversely correlated in colorectal cancer cell lines[23]. We therefore investigated the 5-FU–resistant HCT116 cell line (a widely studied colorectal cancer cell line) to see whether it has accumulation of lipid droplets. To this end, we generated 5-FU–resistant HCT116 (HCT116 FR) cells after prolonged treatment with 5-FU (Fig. 4a). Cell cycle profile analyses confirmed that HCT116 FR cells indeed exhibited less EdU labeling and proliferated more slowly than did parental HCT116 cells (Fig. 4b and Supplementary Fig. 6a). We found that lipid droplet levels were higher in HCT116 FR

cells than in parental cells and that treatment with iDGAT1/2 completely abolished the increases in lipid droplet levels as well as basal lipid droplet levels (Fig. 4c). Furthermore, HCT116 FR cells were more resistant to ferroptosis induced by the GPX4 inhibitor RSL3 or ML162 than were parental HCT116 cells. Importantly, treatment with iDGAT1/2 resensitized HCT116 FR cells to ferroptosis (Fig. 4d, e), suggesting that DGAT-mediated lipid droplet formation drives ferroptosis resistance in HCT116 FR cells.

We also examined the palbociclib-resistant T47D breast cancer cell line (T47D PR), which has been shown to proliferate more slowly than the parental T47D cell line[24]. We confirmed greater palbociclib resistance and a lower proliferation rate in T47D PR cells than in parental T47D cells (Fig. 4f, g). Furthermore, we showed that T47D PR cells had higher lipid droplet levels and more ferroptosis resistance than did parental T47D cells and that ferroptosis resistance in T47D PR cells could be partially reversed by treatment with iDGAT1/2 (Fig. 4h–j).

Finally, radioresistant cells are known be to less proliferative than parental cells[25]. By continually culturing parental H460 cells upon repeated ionizing radiation at 2 Gy, we generated radioresistant H460 (H460 RR) cells (Supplementary Fig. 6b). We found that H460 RR cells exhibited slower growth, higher lipid droplet levels, and greater ferroptosis resistance than did parental H460 cells and that DGAT inhibition partially reversed ferroptosis resistance in H460 RR cells (Supplementary Fig. 6c–f). Collectively, our results from these three slow-cycling therapy-resistant cell line models demonstrated that DGAT-mediated lipid droplet formation in such cancer cells drives ferroptosis resistance.

## 5-FU–resistant xenograft tumors are sensitive to combination treatment with ferroptosis inducers and iDGAT1/2

Using 5-FU–resistant xenograft tumor models, we next studied whether the combination of ferroptosis inducers and iDGAT1/2 is a potential therapeutic strategy for therapy-resistant tumors through induction of ferroptosis in vivo. Because most ferroptosis inducers used in cell culture analyses, including erastin and RSL3, are not suitable for in vivo studies, we used imidazole ketone erastin (IKE), a potent erastin analog suitable for treatment in animals[26], to induce ferroptosis in vivo. As shown in Fig. 4k, l, treatment with IKE significantly reduced the growth of parental HCT116 xenograft tumors, whereas treatment with iDGAT1/2, either alone or in combination with IKE, did not affect tumor growth. In contrast, the combination of IKE and iDGAT1/2, but not either treatment alone, potently suppressed the growth of HCT116 FR tumors (Fig. 4m, n), which is in line with our in vitro data demonstrating that HCT116 FR cells were resistant to ferroptosis and that DGAT inhibition sensitized these cells to ferroptosis (Fig. 4d, e). We confirmed an increase in TAG levels in HCT116 FR tumors compared to the parental tumors; furthermore, this increase in TAG levels was effectively reversed upon treatment with iDGAT1/2

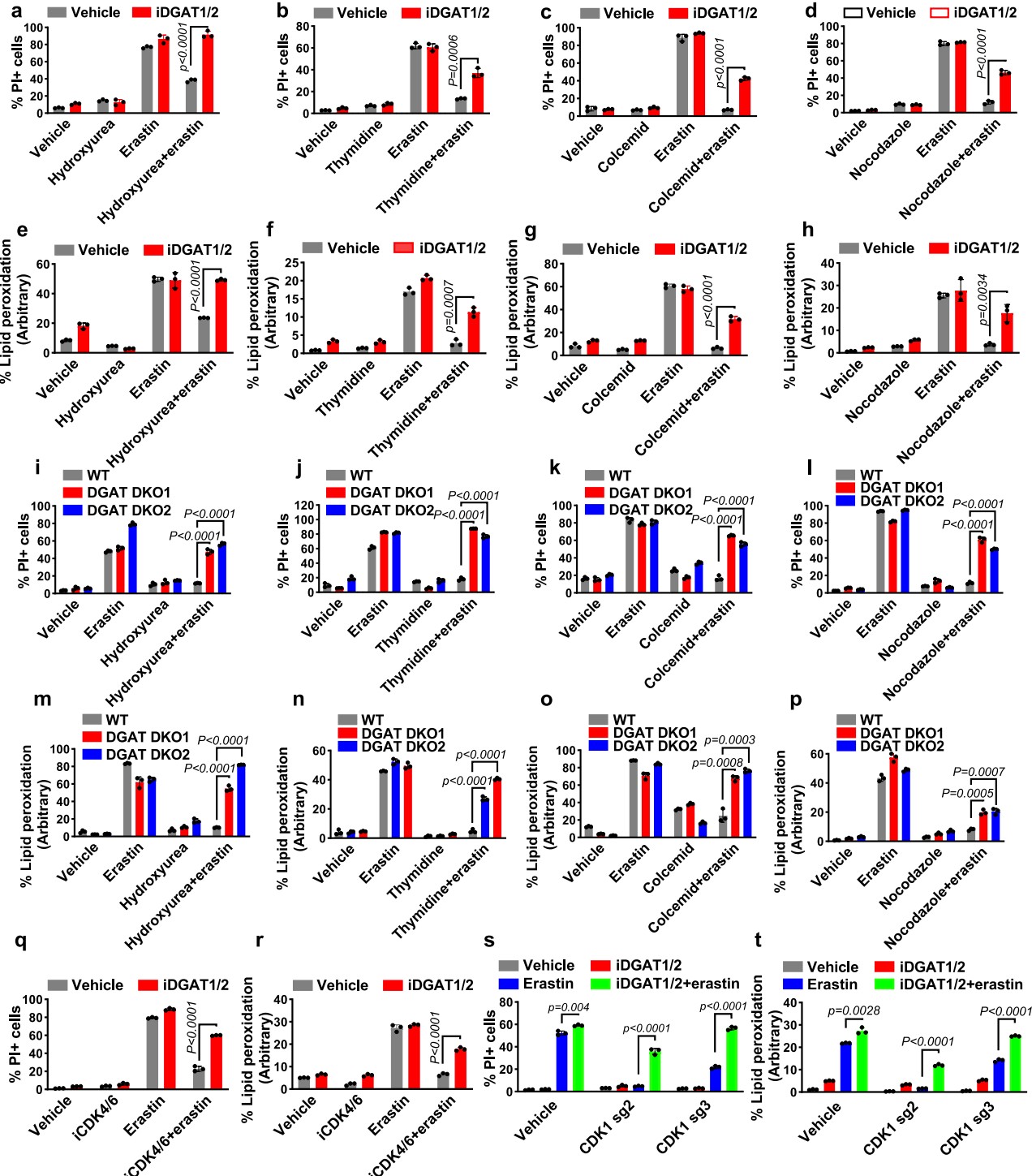

**Fig. 3 | Triacylglyceride formation protects cells from ferroptosis.** Quantification of PI-positive Caki-1 cells treated with 2 µM erastin for 18 h using flow cytometry. Cells were pretreated with a vehicle or iDGAT1/2 with 0.3 mM hydroxyurea (**a**), 1 mM thymidine (**b**), 0.035 µg/ml colcemid (**c**), or 200 nM nocodazole (**d**) for 24 h. Lipid peroxidation measurement in Caki-1 cells treated with 2 µM erastin for 8 h. Cells were pretreated with a vehicle or iDGAT1/2 with 0.3 mM hydroxyurea (**e**), 1 mM thymidine (**f**), 0.035 µg/ml colcemid (**g**), or 200 nM nocodazole (**h**) for 24 h. WT, DGAT DKO1, and DGAT DKO2 Caki-1 cells were pretreated with cell cycle inhibitors for 24 h. Shown are the populations of PI-positive cells after treatment with 2 µM erastin for 18 h (**i**–**l**) and those with lipid peroxidation after treatment with 2 µM erastin for 8 h (**m**–**p**). **q**, **r**, The populations of PI-positive Caki-1 cells pretreated with a vehicle or iDGAT1/2 24 h after treatment with 2 µM erastin for 18 h (**q**) and lipid peroxidation after treatment with 2 µM erastin for 8 h (**r**). **s**, **t** WT, sgCDK1-2, and sgCDK1-3 Caki-1 cells were pretreated with iDGAT1/2 for 24 h. The populations of PI-positive cells after treatment with 2 µM erastin for 18 h (**s**) and lipid peroxidation after treatment with 2 µM erastin for 8 h (**t**) are shown. Mean (± SD) values are shown. $n = 3$. $n$ indicates independent repeats (unpaired, two-tailed $t$-test). Source data are provided as a Source Data file.

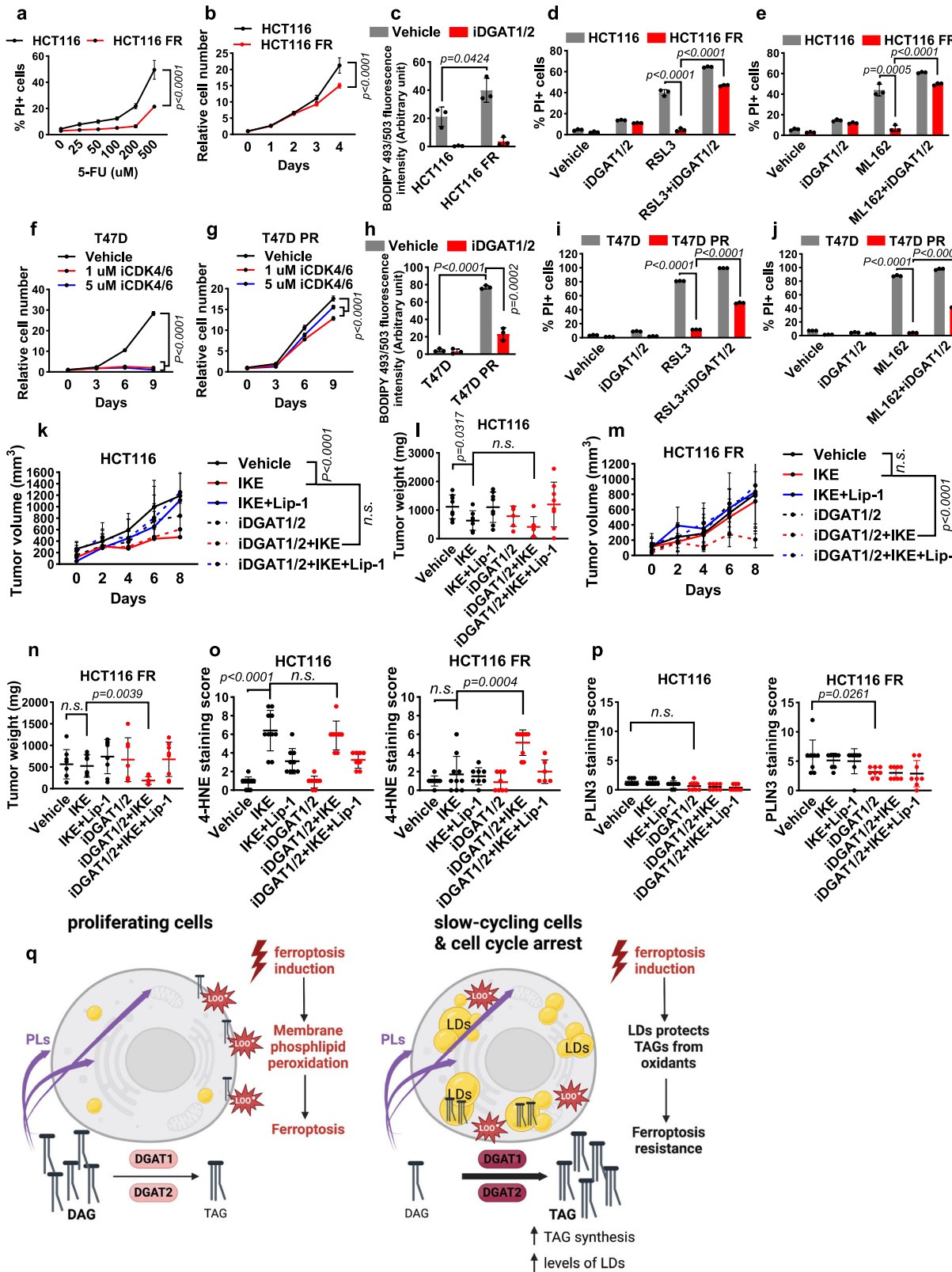

(Supplementary Fig. 6g). (The magnitude of TAG level changes observed in tumor samples was not as pronounced as that seen in cell line studies. This disparity could potentially be attributed to variations in the extent of cell cycle inhibition, with cell cycle inhibitor treatment in cell lines causing much more dramatic suppression of cell proliferation than did 5-FU resistance in tumors.) In both parental and FR xenograft models, treatment with the ferroptosis inhibitor lipiroxstatin-1 restored the growth of tumors treated with IKE alone or in combination with iDGAT1/2 (Fig. 4k–n), confirming that the tumor-suppressive effect of IKE was caused by ferroptosis induction.

Furthermore, immunohistochemical analyses revealed that treatment of parental HCT116 tumors with IKE or combination treatment of HCT116 FR tumors with IKE and iDGAT1/2 significantly increased staining of tumor sections for 4-HNE (a lipid peroxidation

**Fig. 4 | Treatment of slow-cycling therapy-resistant cells. a** PI-positive populations of parental HCT116 and HCT116 FR cells treated with 5-FU for 24 h. **b** HCT116 and HCT116 FR cell growth over 5 days. **c** The relative intensities of BODIPY 493/503 staining in HCT116 and HCT116 FR cells treated with iDGAT1/2 for 48 h. The populations of PI-positive cells treated with 10 μM RSL3 and iDGAT1/2 (**d**) or 10 μM ML162 and iDGAT1/2 (**e**) for 24 h. The relative numbers of parental T47D (**f**) and T47D PR (**g**) cells measured every 3 days. Cells were cultured with or without iCDK4/6. **h** The relative intensities of BODIPY 493/503 staining in T47D and T47D PR cells treated with iDGAT1/2 for 48 h. The populations of PI-positive cells treated with 100 nM RSL3 and iDGAT1/2 for 24 h (**i**) or 200 nM ML162 and iDGAT1/2 for 18 h (**j**). Volumes (**k**) and weights (**l**) of HCT116 xenograft tumors in the indicated treatment groups. $n = 8$ for vehicle, IKE+Lip-1, iDGAT1/2 + IKE+Lip-1; $n = 7$ for IKE,

iDGAT1/2 + IKE; $n = 6$ for iDGAT. Volumes (**m**) and weights (**n**) of HCT116 FR xenograft tumors in the indicated treatments groups, $n = 8$, except for iDGAT ($n = 7$). Immunochemical scoring of 4-HNE (**o**) and PLIN3 (**p**) staining in tumor sections. $n = 8$ in HCT116, except for vehicle, IKE ($n = 10$) and IKE+Lip-1 ($n = 9$). $n = 8$ in HCT116 FR, except for IKE ($n = 10$), iDGAT+IKE ($n = 9$) and iDGAT+IKE+Lip-1 ($n = 6$) were used for 4-HNE and $n = 8$ was used for PLIN3 staining. **q** Working model. See the Discussion for a detailed description. PLs, phospholipids; LOO•, peroxy radical; LD, lipid droplet. Figure 4q is created using BioRender. Mean (±SD) values are shown. $n = 3$. $n$ indicates independent repeats (**a**–**j**), $P$ values were calculated using two-way ANOVA (**a, b, f, g, k, and m**) or an unpaired, two-tailed $t$-test. n.s. not significant. Source data are provided as a Source Data file.

marker) (Fig. 4o and Supplementary Fig. 6j). Of note, in vehicle-treated groups, HCT116 FR tumors exhibited less Ki67 staining than did parental tumors (Supplementary Fig. 6h), which was consistent with the slower cycling of the HCT116 FR cancer cells than corresponding parental cells (Fig. 4b and Supplementary Fig. 6a). We further showed that the average weight of brown adipose tissue samples from xenograft-bearing mice was lower for iDGAT1/2-treated mice than for vehicle-treated mice (Supplementary Fig. 6i), demonstrating that treatment with iDGAT1/2 was effective in vivo. Finally, immunohistochemical staining of tumor sections for PLIN3 (a lipid droplet marker) confirmed higher lipid droplet levels in HCT116 FR tumors than in parental tumors and decreased lipid droplet levels in iDGAT-treated tumors (Fig. 4p and Supplementary Fig. 6k). Collectively, these data demonstrated that slow-cycling therapy-resistant cancer cells exhibit increased lipid droplet levels and ferroptosis resistance, and suggest combining ferroptosis inducers with iDGAT1/2 for treating such therapy-resistant tumors.

## Discussion

During cell proliferation, lipid contents must be doubled (mainly via de novo lipid synthesis and/or lipid uptake from the extracellular environment) in a parental cell before they are partitioned into two daughter cells. Our data suggest that, in cell cycle–arrested cells, lipid uptake from the extracellular milieu continues; consequently, these increased numbers of lipid molecules accumulate in the arrested cells, providing an explanation for the overall increase in lipid content observed in these cells (Fig. 2a). In addition, the increased availability of lipid molecules from the extracellular environment likely contributes to the upregulation of DGAT expression, resulting in further increases in TAG formation. This observation together underscores the significance of lipid uptake in driving the accumulation of TAGs during cell cycle arrest. Future studies will be directed toward elucidating the mechanism by which increased lipid accumulation in arrested cells promotes DGAT expression.

Previous studies demonstrated that TAGs and lipid droplets can either promote[27,28] or suppress[29,30] ferroptosis, depending on the context. In this study, we showed that inactivation of DGATs, pharmacologically or genetically, at least partially reversed the ferroptosis resistance in cell cycle–arrested cells. The present study suggests that in the context of cell cycle arrest, lipid droplets clearly have an anti-ferroptosis role. Importantly, our findings offer mechanistic insights into this antiferroptosis role of lipid droplet formation in cell cycle–arrested cells (but not in normal proliferating cells). Specifically, it is well-established that PUFA-PLs and PUFA-ether PLs provide important substrates for lipid peroxidation[7,15]. Our lipidomic data suggest that in cells arrested in the cell cycle, such as those treated with nocodazole, excessive PUFAs are sequestered within lipid droplets, which provide a protective shield against lipid peroxidation, resulting in ferroptosis resistance in cell cycle–arrested cells (Fig. 4q). Consequently, when lipid droplet formation is blocked through DGAT inhibitor treatment, those PUFAs (which would have been sequestered

in TAGs) are instead incorporated into PLs and ether PLs, which, in turn, renders arrested cells more susceptible to ferroptosis. In contrast, DGAT inhibition did not lead to similar increases in PUFA-PL levels in normal proliferating cells, potentially explaining why DGAT inhibition did not affect ferroptosis sensitivity in such cells.

The role of cell cycle inhibitory proteins in the regulation of ferroptosis have been explored in prior research. For example, it has been demonstrated that p21 (CDKN1A) activation leads to a reduction in the expression of ribonucleotide reductase (RNR) subunits, RRM1 and RRM2, thereby inhibiting nucleotide synthesis. RNR is the rate-limiting enzyme in de novo nucleotide synthesis, and this process occurs in a glutathione-dependent manner. Consequently, p21-mediated inhibition of nucleotide synthesis conserves intracellular glutathione, which, in turn, suppresses ferroptosis[31]. In another recent study, it was shown that the deletion of *p16* (*CDKN2A*) in glioblastomas results in the rewiring of lipid metabolism and renders glioblastoma cells more susceptible to ferroptosis[32]. The observed effects of these two CDK inhibitors in suppressing ferroptosis align with our findings that cell cycle arrest has an inhibitory effect on ferroptosis. However, it is important to note that p21's role in regulating ferroptosis seems to be independent of its classical function in inhibiting CDKs. Furthermore, whether p16's role in regulating lipid metabolism and ferroptosis is connected to its function in CDK inhibition and cell cycle suppression remains unclear. In contrast, our current study emphasizes a general ferroptosis-suppressive effect by cell cycle arrest, irrespective of the specific cell cycle phase at which cells are arrested or specific cell cycle regulatory proteins involved.

Antimitotic drugs kill or suppress the proliferation of cancer cells by inhibiting microtubule formation, chromosome segregation, and/or DNA synthesis in the cells and represent the mainstay of chemotherapy and other cancer therapies[33]. However, resistance to antimitotic drugs is a major challenge in cancer treatment and is at least partly attributed to the slow-cycling nature of such therapy-resistant cancer cells. Leveraging our finding linking cell cycle arrest to lipid droplet formation and ferroptosis resistance, we further demonstrated that slow-cycling cancer cells resistant to palbociclib, 5-FU, or radiation exhibited increased lipid droplet levels and ferroptosis resistance. Our data are consistent with previous studies showing increased lipid droplet and TAG levels in cancer cells treated with other antimitotic drugs (such as nocodazole)[23,34]. Finally, we showed that iDGAT1/2 sensitize therapy-resistant cancer cells to treatment with ferroptosis inducers and synergize with ferroptosis inducers to suppress the growth of 5-FU–resistant xenograft tumors. Therefore, our study may provide a novel therapeutic strategy to overcome resistance to antimitotic drugs in cancer treatment.

## Methods

This research complies with all relevant ethical regulations of The University of Texas MD Anderson Cancer Center, including the Institutional Review Board and Institutional Animal Care and Use Committee.

## Cell culture studies

The Caki-1 (HTB-46), ACHN (CRL-1611), HT1080 (CCL-121), A375 (CRL-1619), 786-O (CRL-1932), HCT116 (CCL-247), H460 (HTB-177), and T47D (HTB-133) cell lines were obtained from the ATCC. The T47D PR cell line was obtained from Dr. Khandan Keyomarsi (MD Anderson Cancer Center)[24]. The TK10 cell line was obtained from Dr. Gordon Mills (MD Anderson Cancer Center). Primary MEFs were established from embryos at embryonic day 13.5 and immortalized via infection with SV40 large T antigen as described previously[14]. All cell lines were cultured in culture media supplemented with 10% (v/v) fetal bovine serum (Gibco, # 26140079) and 1% (v/v) penicillin/streptomycin in a 37 °C incubator with a humidified atmosphere of 20% $O_2$ and 5% $CO_2$. Caki-1 and HCT116 cells were cultured in McCoy's 5a Medium (ATCC, #30-2007). ACHN and HT1080 cells were cultured in Eagle's Minimum Essential Medium (ATCC, #30-2003). A375 cells, TK10 cells, and MEFs were cultured in Dulbecco's modified Eagle's medium (Sigma, #D6429). 786-O, T47D, and H460 cells were cultured in RPMI-1640 medium (Sigma, #R8758). Also, the T47D cell line was cultured with 0.2 U/ml bovine insulin (Sigma, #91077C). All cell lines were cultured in a 10-cm plate and subcultured into a 12-well plate for cell death, lipid peroxidation, and lipid droplet analysis. For a cell viability assay, cells were subcultured into a 96-well plate. Cells were pretreated with cell cycle inhibitors or DGAT inhibitors for 24 h, and then treated with ferroptosis inducers for further assays. Cells were treated with cell cycle inhibitors, hydroxyurea (Sigma, #H8627), thymidine (Sigma, #T1895), colcemid (Sigma, #234109-M), nocodazole (Sigma, #487929-M), or the CDK4/6 inhibitor palbociclib (obtained from Dr. Keyomarsi); the DGAT1 inhibitor T863 (Sigma, #SML0539); the DGAT2 inhibitor PF06427878 (Sigma, #PZ0412); the ferroptosis inducers erastin (Cayman chemical, #17754), RSL3 (Tocris, #6118), or ML162 (Cayman chemical, #20455); the cell death inhibitors ferrostatin-1 (Sigma-Aldrich, #SML0583), necrostatin-1s (BioVision, #2263), or Z-VAD-fmk (R&D Systems, #FMK001); the antioxidant N-acetylcysteine (Sigma, #A9165); or the iron chelator deferoxamine (Sigma, #D9533). See Supplementary Table 1 for detailed information on cell seeding and the drug concentrations used with the cell lines in this study.

## CRISPR/Cas9-mediated gene knockout

Knockout of the *CDK1*, *DGAT1*, *DGAT2*, and *RB1* genes in human cell lines was performed using single guide RNAs (sgRNAs) and CRISPR/Cas9 technology. gRNAs were cloned into the lentiviral plasmid lenti-CRISPR v2 or lentiGuide_Puro (Addgene). sgCDK1: 'TACTTTGTTTCAG GTACCTA', 'GGGTTCCTAGTACTGCAATT', 'CATAAGCACATCCTG AAG AC'; sgDGAT1: 'TTAGAGGCGCCCACCACACC', 'CACCCCCAGGTATGG CATCC', 'CAGGATGCCATACC TGGGGG'; sgDGAT2: 'CGAGTGCGATA CCATTCCCA', 'TATGCAGGACTGGACCACCT', 'CATGTGAACTTGGGAC ACCC'; sgRB1: 'GCTCTGGGTCCTCCTCAGGA', 'TCGCTCACCT GACGA GAGGC', 'TTCATCTGTGGATGGAGTAT'. gRNA clones were transfected into HEK293T cells with psPAX2 packaging plasmid and pMD2.G VSV-G envelope-expressing plasmid as described previously[35,36]. Caki-1 cells were infected with lentivirus along with 0.8 μg/ml hexadimethrine bromide (Polybrene; Millipore, #TR-1003-G) and selected with puromycin (2 μg/ml; InvivoGen, #ant-pr-1) for 3 days. For *CDK1* knockout, gRNAs were cloned into the lentiGuide-Puro vector (Addgene; #52963). Caki-1 cells carrying doxycycline-inducible Cas9 were cultured with tetracycline-free fetal bovine serum (Takara Bio, #631107) and infected with gRNAs targeting *CDK1*. Doxycycline (1 μg/ml; Sigma, D9891) was added into the media for 3 days to generate *CDK1*-knockout cells.

## Cell cycle analysis and EdU flow cytometry assay

Cell cycle analysis was performed using propidium iodide (PI; Roche, #11348639001) staining as described previously[37,38]. Cells were seeded at 50% confluence into 10-cm plates and treated with a vehicle or cell cycle inhibitors for 48 h. Next, cells were collected and fixed in cold 70% ethanol for 4–24 h at −20 °C. Cells were then washed once with phosphate-buffered saline (PBS) and incubated with PI staining solution (50 μg/ml PI, 10 mM Tris/Cl, 5 mM $MgCl_2$, 100 μg/ml RNase) for 30 min at 37 °C. Cell cycle analysis was carried out using an Attune NxT flow cytometer (Thermo Fisher Scientific) with a BL2 detector. At least 5000 single cells were analyzed, and all experiments were performed in triplicate.

To determine cell proliferation in the HCT116 and HCT116 FR cell lines, a Click-iT Plus EdU Alexa Fluor 488 Flow Cytometry Assay Kit (Thermo Fisher Scientific, #C10632) was used according to the manufacturer's protocol. Briefly, $1 \times 10^6$ cells were seeded in a 10-cm plate in triplicate. The next day, cells were labelled with 10 μM EdU for 60 min. After fixation and permeabilization, cells were incubated with 500 μl of Click-iT reaction cocktail for 30 min. EdU-labeled cells were analyzed using the Attune NxT flow cytometer with a BL1 detector.

## Cell death assays

Cell death was analyzed using PI staining with a flow cytometer as described previously[39,40]. Cells were seeded at 70-80% confluence into 12-well plates. The following day, cells were pretreated with a vehicle, cell cycle inhibitors, or iDGAT1/2 for 24 h. Subsequently, cells were treated with ferroptosis inducers. Details on the drug treatment concentrations and times are provided in Supplementary Table 1. To measure cell death, cells (including floating dead cells) were collected and stained with 5 μg/ml PI (Roche, #11348639001), and the percentage of PI-positive dead cells was determined using the Attune NxT flow cytometer with a BL2 detector. At least 5,000 single cells were analyzed per well, and all experiments were performed in triplicate.

## Determination of lipid peroxidation

Lipid peroxidation was examined using BODIPY 581/591 C11 staining as described previously[41–43]. Cells were seeded at 70-80% confluence into 12-well plates. The next day, cells were pretreated with a vehicle, cell cycle inhibitors, or iDGAT1/2 for 24 h. Sunsequently, cells were treated with ferroptosis inducers for 8-24 h. Details on the drug treatment concentrations and times are provided in Supplementary Table 1. Cells were then incubated with 5 μM BODIPY 581/591 C11 (Invitrogen, #D3861) for 30 min at 37 °C and collected. Lipid peroxidation was assessed using the Attune NxT flow cytometer with a BL1 detector. At least 5,000 single cells per well were analyzed.

## Lipid droplet staining

Lipid droplet levels were measured using BODIPY 493/503 staining (Invitrogen, #D3922) as described previously[44]. Cells were seeded at 70-80% confluence into 12-well plates. The following day, cells were treated with a vehicle, cell cycle inhibitors, or iDGAT1/2 and cultured for 48 h. Cells were then incubated with 2 μM BODIPY 493/503 for 15 min at 37 °C and collected. Lipid droplet levels were measured using the Attune NxT flow cytometer with a BL1 detector. At least 5,000 single cells per well were analyzed. For Fig. 2c, live cells were incubated with 2 μM BODIPY 493/503 for 15 min and visualized using an LSM 880 confocal laser scanning microscope (Zeiss).

## Triacylglyceride measurement

Triacylglyceride levels were quantified using the Triglyceride Assay Kit (Abcam, #ab65336) according to the manufacturer's instructions. Briefly, $1 \times 10^7$ Caki-1 cells treated with the indicated cell cycle inhibitors were harvested, washed with cold PBS, and subsequently resuspended in 500 μl of 5% NP-40 solution. After two cycles of heating at 95 °C for 5 min followed by cooling to room temperature to solubilize all triglycerides, the samples were diluted in water and mixed with cholesterol esterase/lipase for 20 min. Then, the triglyceride reaction mix was added and incubated for 60 min at room temperature. The intensity of OD570 nm was measured using a Synergy 2 microplate reader (BioTek). For triacylglyceride measurement in tumor tissues,

the tissues were homogenized in a 5 % NP-40 solution and subsequently diluted in water for further analysis. The concentration of triacylglyceride was normalized to the cell number or the weight of the tumor tissue.

## Cystine uptake assay

Cystine uptake in Caki-1 cells was measured as described previously[45,46]. Briefly, cells were plated in 12-well plates and incubated with a vehicle or cell cycle inhibitors overnight. To measure the cystine uptake, the medium was replaced with fresh Dulbecco's modified Eagle's medium (which contains 200 µM cystine) containing 0.04 µCi [$^{14}$C] Cystine (PerkinElmer, #NEC854010UC), and cells were incubated for 2 h. Cystine uptake was terminated by rapidly rinsing cells twice with cold PBS and lysing them in 0.1 mM NaOH. The radioactivity (in disintegrations per minute) was measured using a TRI-CARB Liquid Scintillation Analyzer (PerkinElmer, #4810TR) in the presence of a quench curve. All experiments were carried out in triplicate.

## Glutathione measurement

Glutathione levels in Caki-1 cells were measured using a GSH-Glo Glutathione Assay (Promega, #V6911) following the manufacturer's instructions and as described previously[47,48]. Briefly, $9 \times 10^3$ Caki-1 cells per well were seeded in a 96-well plate and treated with a vehicle or cell cycle inhibitors for 24 h. The culture medium in the wells was carefully removed, and cells were incubated with 100 µl of GSH-Glo reagent for 30 min at room temperature. Next, 100 µl of reconstituted Luciferin Detection Reagent was added to each well and mixed briefly on a plate shaker. After 15 min, the luminescence was measured using a Synergy 2 microplate reader (BioTek). The result was normalized to the cell numbers in each treatment, and glutathione levels were calculated using a standard curve.

## Labile iron pool measurement

Caki-1 cells were seeded into 12-well plates at a density of $9 \times 10^4$ cells per well and treated with a vehicle or cell cycle inhibitors. The next day, cells were incubated with 20 nM Calcein-AM (Invitrogen, #C3099) for 15 min and washed with PBS. Cells were then harvested and analyzed using the Attune NxT flow cytometer with a BL1 detector.

## Immunoblotting

Immunoblotting was performed as described previously[49,50]. Briefly, cell pellets were lysed using RIPA lysis buffer (Millipore, #20-188), and the protein concentration was determined by a Bradford assay (Bio-Rad, #500-0006) using a NanoDrop 2000 (Thermo Fisher Scientific). For immunoblot analysis, 15-30 µg of protein was used along with antibodies against phospho-Rb ([S780] Cell Signaling Technology, #9307, 1:1,000), RB1 (Cell Signaling Technology, #9309, 1:1,000), CDK1 (Cell Signaling Technology, #9116, 1:1,000), ACSL4 (Santa Cruz Biotechnology, #sc-271800, 1:1,000), GPX4 (R&D Systems, #MAB5457, 1:1,000), SLC7A11 (Cell Signaling Technology, #12691, 1:1,000), FSP1 (Santa Cruz Biotechnology, #sc-377120, 1:300), DHODH (Proteintech, #14877-1-AP, 1:1,000), DGAT1 (Santa Cruz Biotechnology, #sc-271934, 1:300), vinculin (Sigma, #V4505, 1:50,000), and tubulin (Sigma, #T9026, 1:10,000), Goat anti-rabbit IgG secondary antibody (Thermo Scientific, #31460, 1:5,000), Goat anti-mouse IgG secondary antibody (Proteintech, #SA00001-1, 1:5,000). The uncropped scans of the immunoblots used in this study are shown in the Source Data file.

## Quantitative real-time PCR

Real-time PCR was performed as described previously[51,52]. Briefly, total RNA was extracted using TRIzol reagent (Invitrogen, #15596018), and cDNA was synthesized with SuperScript II Reverse Transcriptase (Invitrogen, #18064-014) according to the manufacturer's protocol. qRT-PCR analysis was carried out in triplicate using SYBR GreenER

qPCR SuperMix Universal (Invitrogen, #11762-500) with a Stratagene MX3000P (Agilent Technologies). Primers for DGAT1 (forward, 5′- CT TCTCACTGCCACCTCACA-3′; reverse, 5′-CAGCTGGCATCAGACTGTG T-3′), DGAT2 (forward, 5′-GCTGACCTGGTTCCCATCTA-3′; reverse, 5′-CAGGTGTCGGAGGAGAAGAG-3′) and GAPDH (forward, 5′-CAATGA CCCCTTCATTGACC-3′; reverse, 5′-GATCTCGCTCC TGGAAGATG-3′) were used in this study.

## Lipidomic analyses

Lipidomic analyses were conducted at the Metabolomics Core Facility at MD Anderson Cancer Center. Caki-1 cells were cultured with a vehicle, 200 nM nocodazole, 20 µM iDGAT1/2 (T863, PF06427878), or iDGAT1/2 and nocodazole for 48 h. Approximately 2 million cells were harvested for each run as instructed by the core facility. Briefly, upon harvest, cells were washed with ice cold 0.85% ammonium bicarbonate in deionized water. Cells were then scraped down and pelleted at 400 g for 3 min. After removing the supernatant, cells were snap-frozen in liquid nitrogen and stored at −80 °C before being sent to the facility. To determine the relative abundance of lipid in cells, extracts were prepared and analyzed by the high-resolution mass spectrometry-based lipidomics at the MD Anderson Cancer Center Metabolomics Core Facility. Briefly, for each cell sample, 200 µL of an extraction solution containing 2% Avanti SPLASH® LIPIDOMIX® Mass Spec Standard and 1% 10 mM butylated hydroxytoluene in ethanol was added, and the tubes were vortexed for 10 min. The tubes were placed on ice for 10 min and then centrifuged at 17,000 X g for 10 min at 4 C. The supernatant was transferred to a glass autosampler vial, and the injection volume was 10 µL. Mobile phase A (MPA) consisted of 40:60 acetonitrile: water with 0.1% formic acid and 10 mM ammonium formate, while mobile phase B (MPB) was composed of 90:9:1 iso-propanol:acetonitrile: water with 0.1% formic acid and 10 mM ammonium formate. The chromatographic method included a Thermo Fisher Scientific Accucore C30 column (2.6 µm, 150 × 2.1 mm) maintained at 40 °C, an autosampler tray chilling at 8 °C, a mobile phase flow rate of 0.200 mL/min, and a gradient elution program as follows: 0–3 min, 30% MPB; 3–13 min, 30-43% MPB; 13.1-33 min, 50–70% MPB; 48-55 min, 99% MPB; 55.1–60 min, 30% MPB.

A Thermo Fisher Scientific Orbitrap Fusion Lumos Tribrid mass spectrometer with a heated electrospray ionization source was operated in data dependent acquisition mode, in both positive and negative ionization modes, with scan ranges of 150−1500 $m/z$. An Orbitrap resolution of 240,000 (FWHM) was used for MS1 acquisition, and a spray voltages of 3600 V and 2900 V were used for positive and negative ionization modes, respectively. Vaporizer and ion transfer tube temperatures were set at 275 and 300 °C, respectively. The sheath, auxiliary, and sweep gas pressures were 35, 10, and 0 (arbitrary units), respectively. For MS$^2$ and MS$^3$ fragmentation, a hybridized HCD/CID approach was used. Data were analyzed using Thermo Scientific LipidSearch software (version 5.1) and in-house-written R scripts. The relative abundance of each lipid species was subsequently normalized to the cell count for each treatment condition. The lipidomics data generated in this study have been deposited in the Zenodo database under accession code 10059819. Analyzed lipidomic data are included in Supplementary Data 1.

## Irradiation and clonogenic survival assay

Clonogenic survival assays were performed as described previously[53,54]. H460 and H460 RR cell lines were seeded into 6-well plates at densities of 200-2,000 cells per well in triplicate. The next day, cells were exposed to radiation using an X-Rad320 cabinet irradiator (Precision X-Ray) at doses ranging from 0 to 8 Gy, with a dose rate of 250 MU/min. Fresh medium was added to the plates every 2 days. After 2 weeks, cells were fixed in 10% neutral buffered formalin for 20 min and stained with 0.5% crystal violet (Sigma, #V5265) dissolved in 20% methanol. The number of colonies was divided by the

number of seeded cells and normalized to that of unirradiated control cells.

## Xenograft experiments

The animal experiments were conducted in compliance with a protocol approved by the institutional Animal Care and Use Committee and the Institutional Review Board at MD Anderson Cancer Center. The study adheres to all relevant ethical regulations pertaining to animal research. Mice were housed under specific-pathogen-free conditions with a 12 h light–12 h dark cycle. The ambient temperature was 21–23 °C, with 45% humidity and the mice had ad libitum access to water and standard chow (LabDiet, #5053). Subcutaneous tumor xenograft experiments were conducted as described previously[55,56]. 4-6-week-old female homozygous ($Foxn1^{nu}/Foxn1^{nu}$) nude mice were purchased from the Experimental Radiation Oncology Breeding Core Facility at MD Anderson. HCT116 or HCT116 FR cells ($2.5 \times 10^6$) were mixed with 50% Matrigel (Corning, #354234) and implanted into the left and right flanks of mice subcutaneously. Once visible, tumors were measured, and their volumes were calculated using the equation volume = length × width$^2$ × 0.5. When the tumors reached 200 mm$^3$, the mice were randomly assigned to different treatment groups and given a vehicle, IKE (30 mg/ml; synthesized by Kadmon Corporation), or liproxstatin-1 (10 mg/kg; Sigma, #SML1414) via daily intraperitoneal administration. The DGAT1 inhibitor A-922500 (Selleckchem, #S2674) was dissolved in dimethyl sulfoxide and administered to mice via oral gavage with 1% Tween-80 in water twice a day at 3 mg/kg. Also, the DGAT2 inhibitor PF-06427878 (Sigma, #PZ0412) was dissolved in dimethyl sulfoxide and administered to mice via oral gavage with 1% methylcellulose/80 mM Tris and 0.5% hypromellose acetate succinate twice a day at 2 mg/kg. The maximal tumor burden permitted by the ethics committee is a length of 1.5 cm, and the maximal tumor burden did not exceed the limit.

## Histology and immunohistochemistry

Fresh tumor tissue samples were fixed in 10% neutral buffered formalin overnight, washed once with PBS, and stored in 70% ethanol at 4 °C. The samples were dehydrated and embedded in paraffin according to standard protocols. Embedded tissues were sectioned at a thickness of 5 μm for hematoxylin and eosin as well as 4-HNE, PLIN3, and Ki-67 immunohistochemical staining as described previously[57,58]. The tissue sections were analyzed using an Olympus BX43 microscope. Primary antibodies against 4-HNE (Abcam, #ab46545, 1:400), PLIN3 (Abcam, #ab47638, 1:300), and Ki-67 (Cell Signaling Technology, #9027, 1:400) were incubated overnight at 4 °C. Staining was performed using a VECTASTAIN Elite ABC kit (Vector Laboratories, #PK-6101) and DAB peroxidase substrate kit (Vector Laboratories, #SK-4100). Immunostaining was analyzed using the immunoreactive score[59]. The percentage of positive cells was scored as follows: no stained cells, 0; 1-9% stained cells, 1; 10-50% stained cells, 2; 51-80% stained cells, 3; 81-100% stained cells, 4. Also, the staining intensity was scored as follows: no color reaction, 0; mild reaction, 1; moderate reaction, 2; intense reaction, 3. The final immunoreactive scores were calculated using the formula (score of staining intensity) × (score of percentage of positive cells).

## Statistical analysis and reproducibility

The cell culture experiments were conducted independently at least three times with similar results. Statistical analyses (unpaired, two-tailed $t$-test) shown in bar plots were performed using Prism 9 software (GraphPad Software) with a two-tailed $t$-test. Two-way analysis of variance (ANOVA) was used for grouped analysis. The data are presented as mean ± standard deviations (SD). A $P$ value < 0.05 was considered significant. All immunoblots and immunostaining were repeated three times independently with similar results.

## Reporting summary

Further information on research design is available in the Nature Portfolio Reporting Summary linked to this article.

## Data availability

The lipidomics data generated in this study have been deposited in the Zenodo database under accession code 10059819. All data supporting the findings of this study are available within the paper and Source Data file. Source data are provided with this paper.

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

## Acknowledgements

We thank Donald Norwood from Editing Services, Research Medical Library at The University of Texas MD Anderson Cancer Center for editing the manuscript. This research was supported by the MD Anderson Institutional Research Fund and Bridge Fund; the Emerson Collective Cancer Research Fund; Cancer Prevention & Research Institute of Texas grant RP230072; grants R01CA181196, R01CA244144, R01CA247992, R01CA269646, and U54 CA274220 from the National Institutes of Health; and the N.G. and Helen T. Hawkins Distinguished Professorship for Cancer Research (to B.G.). This research was also supported by the National Institutes of Health Cancer Center Support Grant (P30CA016672) to MD Anderson.

## Author contributions

H.L. performed most of the experiments, with assistance from A.H., J.K.M., G.L., S.D., X.L., L.Z., P.K., and M.L.. K.O. and L.K. conducted the initial lipidomic analyses. M.V.P. provided resources for the initial lipidomic analyses. I.M., B. W., and P.L.L conducted the lipidomic analyses using LC-MS/MS. K.K. provided the T47D PR cell line. B.G. and H.L. designed the experiments. B.G. supervised the project, provided funding support, and established collaboration. B.G. and H.L. wrote the manuscript. All authors commented on the manuscript.

## Competing interests

B.G. reports receiving consultation fees from Guidepoint Global, Cambridge Solutions, and NGM Bio, and is an inventor with patent applications involving targeting ferroptosis in cancer therapy. K.O. and L.K. are former full-time employees of Kadmon Corporation and are now full-time employees of the Carl Icahn Labs and Sanofi, US, respectively. M.V.P. is a former full-time employee of Kadmon Corporation and is now a full-time employee at PMV Pharmaceutics. The other authors declare no competing interests.
