## [Peer Review File · Nature Communications]

REVIEWER COMMENTS

Reviewer #1 (Remarks to the Author):

Ferroptosis is a type of programmed cell death that depends on the presence of iron and is characterized by the accumulation of damaged phospholipids. In this study, the researchers discovered that inducing cell-cycle arrest, by various means, can lead to increased resistance to ferroptosis. Additionally, cancer cell lines that divide slowly and are resistant to drugs also show increased resistance to ferroptosis. A common characteristic of the arrested cells is a significant increase in the quantity of triacylglycerol and lipid droplets. Notably, analyses of the lipids suggest that the fatty acids within the lipid droplets become more unsaturated, and blocking the synthesis of triacylglycerol makes cells more susceptible to ferroptosis.

Based on these findings, the authors propose a model in which lipid droplets play a role in promoting ferroptosis resistance in cell cycle arrested cells. It is worth noting that lipid droplets tend to accumulate in cell cycle arrested cells and have been linked to protecting cancer cells from ferroptosis. However, our understanding of the relationship between lipid droplets and ferroptosis is still in its early stages. Although the observations and proposed model are intriguing, the lack of mechanistic data linking cell cycle arrest to the biogenesis of lipid droplets and the protective effects of lipid droplets in cell cycle arrested cancer cells leaves important questions unanswered. Further studies to address these mechanistic questions would provide valuable support for the proposed model.

1) One of the most pressing questions that remains unanswered is how lipid droplets contribute to ferroptosis resistance in this context. It is clear that lipid droplets play distinct roles in regulating lipid damage under different conditions. How does blocking lipid droplets lead to resensitization of cells to ferroptosis? Is this due to an effect on the amount of unsaturation of the phospholipid membranes? There are many possibilities that could explain the observations.

2) The lipidomics data presented in this manuscript are insufficient to support the findings due to a lack of acceptable methods used for their identification. It appears that the lipids were identified solely based on their m/z values, which falls short of the minimum requirements to identify lipids from LC-MS data. To identify a lipid, tandem mass spectrometry, involving fragmentation during MS/MS scans, must be employed. If LC-MS/MS was employed, please update the methods accordingly to indicate the details of the analysis. Alternatively, these experiments should be repeated using appropriate methods.

3) The mechanism underlying the increase in lipid droplets following cell cycle arrest is not clear. The proposed upregulation of DGATs appears to be highly speculative and unlikely to be responsible for the

observed changes. Further research is needed to determine the actual mechanism driving the increase in lipid droplets in response to cell cycle arrest.

Reviewer #2 (Remarks to the Author):

The author aims to study the effect of cell Inhibitors along with ferroptosis inducers in slow cycling antimitotic drug-resistant cancer cells and established a ferroptosis-inducing therapeutic strategy to target slow-cycling therapy to overcome resistance to antimitotic drugs in cancer treatment. It is a novel work for the cancer research area. However, a few points need to be verified and the detailed comments are mentioned below.

- The Animal Ethical Committee approval details are not available for the in-vivo work on mice.
- The authors assigned lipid species and FA chain composition using only MS instead of MS/MS spectra. However, MS is not sufficient to analyze lipid species in biological mixtures. Further details on how the authors identified lipid species by MS or MS/MS, and the normalization of data should be provided.
- PC and SM are difficult to distinguish by mass spectrometry as they have the same characteristic fragment ion at m/z 184 [C₅H₁₅O₄NP]⁺ for choline. SM has additional fragment ions at m/z 264 and 282 for sphingosine. The authors should discuss how they clearly identify PC and SM.
- The figures (2A and G) demonstrate a significant increase in most of the lipids in the experimental group treated with nocodazole. This phenomenon is unusual. However, it should be noted that the authors extracted and compared total lipids from cells, without isolating lipids from the enriched lipid droplet. Are there other reports that lipid expression is increased when the cell cycle is arrested in G2/M phase? Additionally, a more detailed explanation regarding potential sources of experimental errors in lipid identification and quantification would be helpful.
- In conclusion, the topic is interesting and important, but the manuscript requires data to reinforce the statements.

Reviewer #3 (Remarks to the Author):

It has been known for some while that under different stress conditions lipid droplets and TAGs accumulate in cells in vitro. Studies have also demonstrated that under cell cycle arrest both lipid droplets and TAGs accumulate consistent with was the authors show in their experiments. It is essential

that all the relevant references are included in the manuscript to avoid misrepresentation of the novelty of the findings. For instance, careful review of lipid droplet and tag accumulation in cell cycle arrest and in cell death needs to be done and these previous findings needs to be incorporated into the manuscript. With these previous work in mind, what this paper adds to the existing knowledge is that lipid droplet accumulation and cell cycle arrest has our role in ferroptosis resistance. The experimental design is done very carefully, hypotheses are tested in multiple cell lines using different models to induce cell cycle, so I have no concerns about the robustness after study design, the findings and the reproducibility. However, I believe the novelty of the findings is somewhat incremental for Nature Communications. Below are some comments about the existing experiments and the missing pieces to be included to fully support the conclusions being made.

Define what “Rb” is in the second paragraph of the results section. What about the role of p21 or p16? What is the effect of their overexpression in ferroptosis and resistance?

Figure 2A reports lipidomic results after not quite as all treatment. The results show that all lipids accumulate regardless of the lipid class which indicates upregulation of lipid per synthesis at the global level. The authors focus on TAGs and provide some functional involvement by using DGAT1/2 inhibitors which can cause modeling at the phospholipid level (due to the decreased TAG accumulation), which could be responsible for differences in resistance. These should, at least, be discussed.

In figure S3 the expression level of they get one and two are investigated however it is unclear whether protein or transcript levels are reported for DGAT2. The western data four they get one needs to be quantified using biological replicates.

The authors use lipid droplet levels as a common end point assay. The lipid droplet data needs to be integrated with TAG measurements to support the discussions on the effects on lipid droplet formation and TAG synthesis.

It would be interesting to see the lipidomics results from the xenograft models treated with ferroptosis inducers and TAG inhibitors and see whether there is a correlation between the phenotype and lipid changes in vivo as well.

Finally, it will be important for the authors to address the effects of the inhibition DGAT1/2 inhibition at the lipid levels. While the TAG formation would be suppressed, what happens to the rest of the lipidome? Where do the lipid production gets challenged or does the de novo synthesis gets shut down?

Detailed point-by-point response to the reviewers' comments:

Note to reviewers: We thank the reviewers for their valuable time and effort dedicated to reviewing our manuscript. Their insightful comments have played a critical role in enhancing the quality of our work. Please find below our detailed point-by-point response addressing all the comments raised by the reviewers. In order to facilitate the review process for the reviewers, we have incorporated all the new data as figures within this rebuttal letter, directly referring to the corresponding figures and text in our revised manuscript. In addition, we have used colored text to clearly highlight the revisions made in our manuscript.

REVIEWER COMMENTS

Reviewer #1 (Remarks to the Author):

Ferroptosis is a type of programmed cell death that depends on the presence of iron and is characterized by the accumulation of damaged phospholipids. In this study, the researchers discovered that inducing cell-cycle arrest, by various means, can lead to increased resistance to ferroptosis. Additionally, cancer cell lines that divide slowly and are resistant to drugs also show increased resistance to ferroptosis. A common characteristic of the arrested cells is a significant increase in the quantity of triacylglycerol and lipid droplets. Notably, analyses of the lipids suggest that the fatty acids within the lipid droplets become more unsaturated, and blocking the synthesis of triacylglycerol makes cells more susceptible to ferroptosis.

Based on these findings, the authors propose a model in which lipid droplets play a role in promoting ferroptosis resistance in cell cycle arrested cells. It is worth noting that lipid droplets tend to accumulate in cell cycle arrested cells and have been linked to protecting cancer cells from ferroptosis. However, our understanding of the relationship between lipid droplets and ferroptosis is still in its early stages. Although the observations and proposed model are intriguing, the lack of mechanistic data linking cell cycle arrest to the biogenesis of lipid droplets and the protective effects of lipid droplets in cell cycle arrested cancer cells leaves important questions unanswered. Further studies to address these mechanistic questions would provide valuable support for the proposed model.

1) One of the most pressing questions that remains unanswered is how lipid droplets contribute to ferroptosis resistance in this context. It is clear that lipid droplets play distinct roles in regulating lipid damage under different conditions. How does blocking lipid droplets lead to resensitization of cells to ferroptosis? Is this due to an effect on the amount of unsaturation of the phospholipid membranes? There are many possibilities that could explain the observations.

2) The lipidomics data presented in this manuscript are insufficient to support the findings due to a lack of acceptable methods used for their identification. It appears that the lipids were identified solely based on their m/z values, which falls short of the minimum requirements to identify lipids from LC-MS data. To identify a lipid, tandem mass spectrometry, involving fragmentation during MS/MS scans, must be employed. If LC-MS/MS was employed, please update the methods accordingly to indicate the details of the analysis. Alternatively, these experiments should be repeated using appropriate methods.

We greatly appreciate the thorough evaluation and insightful comments provided by the reviewer. Since these two questions are related to each other, we address them together below.

In response to the concern raised by this reviewer in the second question regarding the use of LC-MS in our original lipidomic analysis, we have repeated our lipidomic experiments using LC-MS/MS (as detailed in the revised Method section; see colored text in pages 24-25). Correspondingly, we have replaced all lipidomic data presented in Fig. 2.

Our new lipidomic results confirmed our original observation that treatment with nocodazole resulted in a global increase in the relative abundance of many lipid species, with the increase in triacylglycerols (TAGs) and ether-TAGs (TAG-Os) being most significant (**Rebuttal Letter Figure 1**; Fig. 2a, b in the revised manuscript).

Figure 1. Comparative analysis of lipid species in nocodazole-treated vs vehicle-treated Caki-1 cells shown as a volcano plot (A) and comparison between non-TAG vs. TAG groups (B). **: P<0.0001.**

In addition, compared to nocodazole treatment alone, the combination treatment with DGAT inhibitors (iDGAT1/2) and nocodazole reduced the levels of TAG and TAG-O but increased diacylglycerol (DAG) levels, as expected; interestingly, DGAT inhibitor treatment also increased the levels of many phospholipids (PLs) (**Rebuttal Letter Figure 2A**; Fig. 2j in the revised manuscript). A more detailed analysis focusing on PLs revealed a substantial increase in the levels of multiple PUFA-PLs and PUFA-ether PLs (particularly in the form of phosphoethanolamines [PEs]) following DGAT inhibitor treatment (**Rebuttal Letter Figure 2B**; Fig. 2l in the revised manuscript). (Of note, our prior lipidomic analysis conducted with LC-MS failed to uncover this particular finding.)

Our new findings therefore offer valuable mechanistic insights that address the first question raised by the reviewer: how blocking lipid droplets leads to resensitization of cells to ferroptosis.

Figure 2. Volcano plots comparing all lipid species (A) or phospholipids (B) between iDGAT1/2 + nocodazole and nocodazole treatment conditions in Caki-1 cells.

Specifically, it is well-established that PUFA-PLs and PUFA-ether PLs provide important substrates for lipid peroxidation (Kagan et al., 2017; Zou et al., 2020). Our new data suggest that in cells arrested in the cell cycle, such as those treated with nocodazole, excessive PUFAs are sequestered within lipid droplets, providing a protective shield against lipid peroxidation and ferroptosis. Consequently, when lipid droplet formation is blocked through DGAT inhibitor treatment, those PUFAs (which would have been sequestered in TAGs) are instead incorporated into PLs and ether PLs, which, in turn, renders arrested cells more susceptible to ferroptosis. We have incorporated this discussion in our revised manuscript (see colored text in the 2nd paragraph of page 14 to page 15).

3) The mechanism underlying the increase in lipid droplets following cell cycle arrest is not clear. The proposed upregulation of DGATs appears to be highly speculative and unlikely to be responsible for the observed changes. Further research is needed to determine the actual mechanism driving the increase in lipid droplets in response to cell cycle arrest.

To address this reviewer’s comment, we conducted additional experiments and generated new data to gain further mechanistic insights (**Rebuttal Letter Figure 3**; Supplementary Fig. 5 in the revised manuscript). Our new data demonstrate that, when cells were cultured in lipoprotein-free fetal bovine serum (FBS), the accumulation of lipid droplets (**Panels A, B**), the increase in DGAT expression (**Panels C, D**), and the development of resistance to ferroptosis (**Panels E, F**) induced by cell cycle arrest were largely abolished. This finding strongly suggests that the observed phenotypes related to DGAT, lipid droplet formation, and ferroptosis resistance in cell cycle-arrested cells primarily depend on the uptake of lipids from the extracellular environment.

Figure 3. Lipid droplet accumulation (A, B), DGAT1 and 2 expression (C, D), and ferroptosis inhibition (E, F) require serum lipoproteins. R-FBS, regular-fetal bovine serum; LF-FBS, lipoprotein free-fetal bovine serum. *: P<0.05, **: P<0.01, ***: P<0.001, ****: P<0.0001

During cell proliferation, lipid contents must be doubled (by de novo lipid synthesis and/or lipid uptake from the extracellular environment) before they are partitioned into two daughter cells. Our data suggest that, in cell cycle-arrested cells, lipid uptake from the extracellular milieu continues; consequently, these increased numbers of lipid molecules accumulate in the arrested cells, providing an explanation for the overall increase in lipid content observed in these cells. In addition, the increased availability of lipid molecules from the extracellular environment likely contributes to the upregulation of DGAT expression, resulting in further increases in TAG formation. This observation together underscores the significance of lipid uptake in driving the accumulation of TAGs during cell cycle arrest. While we acknowledge that the precise mechanism by which increased lipid accumulation in arrested cells promotes DGAT expression remains to be fully elucidated, we believe that our new data have provided additional insights into the underlying mechanisms responsible for lipid droplet accumulation during cell cycle arrest. We have now incorporated this discussion into our revised manuscript (see colored text in the first paragraph on page 14). We hope that these findings address the reviewer's concerns and enhance the comprehensiveness of our study.

Reviewer #2 (Remarks to the Author):

The author aims to study the effect of cell Inhibitors along with ferroptosis inducers in slow cycling antimitotic drug-resistant cancer cells and established a ferroptosis-inducing therapeutic strategy to target slow-cycling therapy to overcome resistance to antimitotic drugs in cancer treatment. It is a novel work for the cancer research area. However, a few points need to be verified and the detailed comments are mentioned below.

We thank the reviewer for highlighting the novelty and importance of our study. In the following sections, we have thoroughly addressed the additional technical comments provided by this reviewer.

- *The Animal Ethical Committee approval details are not available for the in-vivo work on mice.*

We thank the reviewer for pointing this out. In the revised manuscript, we have included the following sentences under the Method section “The animal experiments were conducted in compliance with a protocol approved by the institutional Animal Care and Use Committee and the Institutional Review Board at MD Anderson Cancer Center. The study adheres to all relevant ethical regulations pertaining to animal research.”

- *The authors assigned lipid species and FA chain composition using only MS instead of MS/MS spectra. However, MS is not sufficient to analyze lipid species in biological mixtures. Further details on how the authors identified lipid species by MS or MS/MS, and the normalization of data should be provided.*

In response to the concern raised by this reviewer (as well as reviewer 1), we have repeated our lipidomic experiments using LC-MS/MS. Our new lipidomic results confirmed our original observation that treatment with nocodazole resulted in a global increase in the relative abundance of many lipid species, with the increase in triacylglycerols (TAGs) and ether TAGs (TAG-Os) being most significant (**Rebuttal Letter Figure 4**; Fig. 2a, b in the revised manuscript).

In addition, compared to nocodazole treatment alone, the combination treatment with DGAT inhibitors (iDGAT1/2) and nocodazole reduced the levels of TAG and TAG-O but increased diacylglycerol (DAG) levels, as expected; interestingly, DGAT inhibitor treatment also increased the levels of many phospholipids (PLs) (**Rebuttal Letter Figure 5A**; Fig. 2j in the revised manuscript). A more detailed analysis focusing on PLs revealed a substantial increase in the levels of multiple PUFA-PLs and PUFA-ether PLs (particularly in the form of phosphoethanolamines [PEs]) following DGAT inhibitor treatment (**Rebuttal Letter Figure 5B**; Fig. 2l in the revised manuscript). (Of note, our prior lipidomic analysis conducted with LC-MS failed to uncover this particular finding.)

Figure 4. Comparative analysis of lipid species in nocodazole-treated vs vehicle-treated Caki-1 cells shown as a volcano plot (A) and comparison between non-TAG vs. TAG groups (B). **: P<0.0001.**

Our new findings also offer valuable mechanistic insights into the process by which inhibiting lipid droplet formation can restore the sensitivity of cell cycle-arrested cells to ferroptosis. Specifically, it is well-established that PUFA-PLs and PUFA-ether PLs provide important substrates for lipid peroxidation (Kagan et al., 2017; Zou et al., 2020). Our new data suggest that in cells arrested in the cell cycle, such as those treated with nocodazole, excessive PUFAs are sequestered within lipid droplets, providing a protective shield against lipid peroxidation and ferroptosis. Consequently, when lipid droplet formation is blocked through DGAT inhibitor treatment, those PUFAs (which would have been sequestered in TAGs) are instead incorporated into PLs and ether PLs, which, in turn, renders arrested cells more susceptible to ferroptosis.

Figure 5. Volcano plots comparing all lipid species (A) or phospholipids (B) between iDGAT1/2 + nocodazole and nocodazole treatment conditions in Caki-1 cells.

Collectively, the new data have not only validated our original findings but also yielded fresh and valuable mechanistic insights. Correspondingly, we have updated Fig. 2 and provided a detailed methodology description of our lipidomic analyses in the Method section in the revised manuscript.

• PC and SM are difficult to distinguish by mass spectrometry as they have the same characteristic fragment ion at m/z 184 [C₅H₁₅O₄NP]⁺ for choline. SM has additional fragment ions at m/z 264 and 282 for sphingosine. The authors should discuss how they clearly identify PC and SM.

We acknowledge the challenge of distinguishing between PC and SM using mass spectrometry in the absence of fragmentation. In our revised manuscript, we have employed ultra-high resolution, accurate mass LC-MS/MS. This allowed us to confirm the presence of additional fragment ions at m/z 264 and 282, which correspond to sphingosine. These identifications were facilitated by the Lipid Search software version 5.0 from Thermo Fisher Scientific. A detailed explanation of the analytical and annotation workflows has been added to the Methods section in the revised manuscript.

• The figures (2A and G) demonstrate a significant increase in most of the lipids in the experimental group treated with nocodazole. This phenomenon is unusual. However, it should be noted that the authors extracted and compared total lipids from cells, without isolating lipids from the enriched lipid droplet. Are there other reports that lipid expression is increased when the cell cycle is arrested in G2/M phase? Additionally, a more detailed explanation regarding potential sources of experimental errors in lipid identification and quantification would be helpful.

This reviewer correctly pointed out that, in our study, we isolated lipids from whole cells rather than specifically from lipid droplets. In our updated lipidomic analysis utilizing LC-MS/MS, we once again observed an increase in most lipid species following nocodazole treatment (see **Rebuttal Letter Figure 4**). The phenomenon of increased lipid profiles in cells arrested in G2/M phase was indeed reported previously (Wong et al., 2018) (which was cited in our manuscript). In this study, U2OS cells treated with nocodazole exhibited a significant accumulation of lipid droplets, and the lipidomic profile revealed a substantial upregulation of lipid species, particularly in the levels of TAG and phospholipids.

• In conclusion, the topic is interesting and important, but the manuscript requires data to reinforce the statements.

We are grateful for the thoughtful and positive feedback provided by this reviewer. We hope this reviewer will agree that the additional data we have generated in response to the constructive comments from this reviewer and others have significantly bolstered the robustness of our conclusions.

Reviewer #3 (Remarks to the Author):

It has been known for some while that under different stress conditions lipid droplets and TAGs accumulate in cells in vitro. Studies have also demonstrated that under cell cycle arrest both lipid droplets and TAGs accumulate consistent with was the authors show in their experiments. It is essential that all the relevant references are included in the manuscript to avoid misrepresentation of the novelty of the findings. For instance, careful review of lipid droplet and tag accumulation in cell cycle arrest and in cell death needs to be done and these previous findings needs to be incorporated into the manuscript. With these previous work in mind, what this paper adds to the existing knowledge is that lipid droplet accumulation and cell cycle arrest has our role in

ferroptosis resistance. The experimental design is done very carefully, hypotheses are tested in multiple cell lines using different models to induce cell cycle, so I have no concerns about the robustness after study design, the findings and the reproducibility. However, I believe the novelty of the findings is somewhat incremental for Nature Communications. Below are some comments about the existing experiments and the missing pieces to be included to fully support the conclusions being made.

We thank the reviewer for the meticulous and insightful evaluation of our work and appreciate his/her positive comment on the robustness and reproducibility of our findings. We also kindly ask the reviewer to consider the following points while evaluating the conceptual novelty of our study

In this study, we have unveiled a hitherto unrecognized phenomenon where cell cycle arrest exerts a suppressive effect on ferroptosis, irrespective of the specific cell cycle phase at which cells are arrested. This discovery is particularly intriguing given that cell cycle arrest typically coincides with the induction of cell death, rendering the observed suppressive effect of cell cycle arrest on ferroptosis both surprising and novel. Mechanistically, our comprehensive lipidomic and functional studies revealed that cell cycle arrest suppresses ferroptosis by promoting DGAT-dependent triglyceride (TAG) and lipid droplet formation. Finally, leveraging our mechanistic understanding of cell cycle arrest and ferroptosis, we further showed that combining ferroptosis inducers with DGAT inhibitors overcomes ferroptosis resistance and suppresses the growth of therapy-resistant slow-cycling tumors.

For the second point, this reviewer has raised the concern that our findings might be perceived as incremental, citing previous studies demonstrating an increase in TAG and lipid droplet levels during cell cycle arrest. We have already cited the relevant publications (such as (Wong et al., 2018)) in our previous manuscript and further clarified this point in the revised manuscript (see colored text in page 16). However, we wish to emphasize that the prior observation that cell cycle arrest leads to an elevation in TAG and lipid droplet levels by itself does not predict or explain the ferroptosis-suppressive effect of cell cycle arrest. This is due to the complex and context-dependent role of lipid droplets in ferroptosis, as mentioned in our manuscript. Additionally, it is important to consider that cell cycle arrest can trigger numerous downstream effects capable of influencing ferroptosis sensitivity, positively or negatively. Therefore, our observation that cell cycle arrest had a potent suppressive effect on ferroptosis is indeed intriguing.

In light of these considerations, we argue that our principal findings are not merely incremental but reveal novel mechanistic insights into how cell cycle arrest suppresses ferroptosis. We hope the reviewer will concur that these conceptual advancements warrant its publication in *Nature Communications*. In the following sections, we have thoroughly addressed the additional technical comments provided by this reviewer.

Define what “Rb” is in the second paragraph of the results section. What about the role of p21 or p16? What is the effect of their overexpression in ferroptosis and resistance?

Rb refers to retinoblastoma protein. CDK4/6-mediated phosphorylation of Rb protein is a critical step in S-phase entry. Rb knockout cells were used in our study to determine whether the effect of CDK4/6 inhibition on ferroptosis suppression depends on Rb-mediated S-phase entry. Our data

showed that Rb deficiency largely abrogated CDK4/6 inhibition-induced cell-cycle arrest and ferroptosis suppression (see Fig. 1l, m in the manuscript), thereby suggesting that an intact Rb checkpoint is required for ferroptosis suppression induced by CDK4/6 inhibition. This result further established that the effect of CDK4/6 inhibition on ferroptosis suppression is mediated through its function in inducing cell cycle arrest. We have now defined Rb and made these points more clearly in the revised manuscript (see colored text in the first paragraph on page 6).

The roles of p21 (CDKN1A) and p16 (CDKN2A) in the regulation of ferroptosis have been explored in prior research. For example, it has been demonstrated that p21 activation leads to a reduction in the expression of ribonucleotide reductase (RNR) subunits, RRM1 and RRM2, thereby inhibiting nucleotide synthesis. RNR is the rate-limiting enzyme in de novo nucleotide synthesis, and this process occurs in a glutathione-dependent manner. Consequently, p21-mediated inhibition of nucleotide synthesis conserves intracellular glutathione levels, which, in turn, suppresses ferroptosis (Tarangelo et al., 2022). In another recent study, it was shown that the deletion of p16 in glioblastomas results in the rewiring of lipid metabolism and renders glioblastoma cells more susceptible to ferroptosis (Minami et al., 2023).

The observed effects of these two CDK inhibitors in suppressing ferroptosis align with our findings that cell cycle arrest has an inhibitory effect on ferroptosis. However, it is important to note that p21's role in regulating ferroptosis seems to be independent of its classical function in inhibiting CDKs. Furthermore, whether p16's role in regulating lipid metabolism and ferroptosis is connected to its function in CDK inhibition and cell cycle suppression remains uncertain. Given these considerations and the fact that our focus is on examining the general impact of cell cycle arrest on ferroptosis, we did not investigate the roles of p21 and p16 in this manuscript. Nevertheless, we acknowledge the relevance of these studies to our current findings and have included a discussion on these points in our revised manuscript (see colored text in the 2nd paragraph on page 15).

Figure 2A reports lipidomic results after not quite as all treatment. The results show that all lipids accumulate regardless of the lipid class which indicates upregulation of lipid per synthesis at the global level. The authors focus on TAGs and provide some functional involvement by using DGAT1/2 inhibitors which can cause modeling at the phospholipid level (due to the decreased TAG accumulation), which could be responsible for differences in resistance. These should, at least, be discussed.

In our previous manuscript, we performed lipidomic analyses using LC-MS. In response to the concern raised by other reviewers, we have repeated our lipidomic experiments using LC-MS/MS (which allows more precise analyses of lipid structure). Our new lipidomic results confirmed our original observation revealing an overall

Figure 6. Comparative analysis of lipid species in nocodazole-treated vs vehicle-treated Caki-1 cells shown as a volcano plot (A) and comparison between non-TAG vs. TAG groups (B). **: P<0.0001.**

increase in lipid levels following nocodazole treatment, with the increase in triacylglycerols (TAGs) and ether-TAGs (TAG-Os) being most significant (**Rebuttal Letter Figure 6**; Fig. 2a, b in the revised manuscript).

In addition, compared to nocodazole treatment alone, the combination treatment with DGAT inhibitors (iDGAT1/2) and nocodazole reduced the levels of TAG and TAG-O but increased diacylglycerol (DAG) levels, as expected; interestingly, DGAT inhibitor treatment also increased the levels of many phospholipids (PLs) (**Rebuttal Letter Figure 7A**; Fig. 2j in the revised manuscript). A more detailed analysis focusing on PLs revealed a substantial increase in the levels of multiple PUFA-PLs and PUFA-ether PLs (particularly in the form of phosphoethanolamines [PEs]) following DGAT inhibitor treatment (**Rebuttal Letter Figure 7B**; Fig. 2l in the revised manuscript). (Of note, our prior lipidomic analysis conducted with LC-MS failed to uncover this particular finding.)

Figure 7. Volcano plots comparing all lipid species (A) or phospholipids (B) between iDGAT1/2+nocodazole and nocodazole treatment conditions in Caki-1 cells.

Importantly, these new findings offer valuable mechanistic insights into the process by which inhibiting lipid droplet formation can restore the sensitivity of cell cycle-arrested cells to ferroptosis. Specifically, it is well-established that PUFA-PLs and PUFA-ether PLs provide important substrates for lipid peroxidation (Kagan et al., 2017; Zou et al., 2020). Our new data suggest that in cells arrested in the cell cycle, such as those treated with nocodazole, excessive PUFAs are sequestered within lipid droplets, providing a protective shield against lipid peroxidation and ferroptosis. Consequently, when lipid droplet formation is blocked through DGAT inhibitor treatment, those PUFAs (which would have been sequestered in TAGs) are instead incorporated into PLs and ether PLs, which, in turn, renders arrested cells more susceptible to ferroptosis. We have incorporated this discussion in our revised manuscript (see colored text in the 2nd paragraph on page 14).

In figure S3 the expression level of they get one and two are investigated however it is unclear whether protein or transcript levels are reported for DGAT2. The western data four they get one needs to be quantified using biological replicates.

In our manuscript, we have shown that cell cycle inhibition increased DGAT2 mRNA levels (**Rebuttal Letter Figure 8A**; Supplementary Fig. 3g in the revised manuscript). However, our efforts to detect endogenous DGAT2 protein levels, in line with findings from other studies (Brandt et al., 2016), proved unsuccessful. This challenge may be related to previous observations

showing that DGAT2 protein is highly unstable and subjects to rapid degradation by proteasomes via an ubiquitin-dependent mechanism (Brandt et al., 2016; Choi et al., 2014). We have now added this point in the revised manuscript (see colored text in the 2nd paragraph, page 8).

Following the thoughtful suggestion from this reviewer, we conducted Western blot experiments for DGAT1 for at least three times. As part of our revisions, we have now incorporated a representative Western blot image for DGAT1 and included a bar graph displaying the averaged DGAT1 protein levels along with statistical analyses in the revised manuscript (**Rebuttal Letter Figure 8B, C**; Supplementary Fig. 3f in the revised manuscript).

Figure 8. The expression levels of DGAT1 and DGAT2 upon cell cycle inhibitor treatment in Caki-1 cells. (A) Relative mRNA levels of DGAT2 under indicated treatments. **(B)** Representative western blots of DGAT1 under indicated treatments. **(B)** The average relative DGAT1 protein levels normalized to vinculin protein levels from three independent experiments. *: P<0.05, **: P<0.01, ***: P<0.001.

The authors use lipid droplet levels as a common end point assay. The lipid droplet data needs to be integrated with TAG measurements to support the discussions on the effects on lipid droplet formation and TAG synthesis.

To address this question from the reviewer, we quantified TAG levels under a range of cell cycle inhibition conditions, including treatment with various cell cycle inhibitors, CDK4/6 inhibitors (iCDK4/6), and the genetic deletion of CDK1. We employed the Triglyceride Assay Kit-Quantification (Abcam, 65336) for this analysis. As shown in **Rebuttal Letter Figure 9** (Figure 2g in the revised manuscript), our findings confirm a consistent elevation in TAG levels across these different cell cycle inhibition conditions.

Figure 9. The levels of triglyceride (TAG) in cells with cell cycle inhibition. *: P<0.001, ****: P<0.0001.**

It would be interesting to see the lipidomics results from the xenograft models treated with ferroptosis inducers and TAG inhibitors and see whether there is a correlation between the phenotype and lipid changes in vivo as well.

We agree with the reviewer's suggestion regarding the potential value of conducting lipidomic analyses on xenograft tumors. However, we faced several challenges that impeded the execution of these lipidomic experiments during the limited time frame for our manuscript revision, primarily

due to the exceedingly high demand for services in our metabolomic core facility and the inherent technical complexities associated with lipidomic analyses of tumor samples.

As an alternative approach to validate our findings in an in vivo context, we conducted measurements of TAG levels in these xenograft tumor samples. As shown in **Rebuttal Letter Figure 10** (Supplementary Fig. 6g) in the revised manuscript), our results confirmed an increase in TAG levels in 5-FU resistant HCT116 xenograft tumors compared to the parental tumors. Furthermore, this increase in TAG levels was effectively reversed upon treatment with DGAT1/2 inhibitors (iDGAT1/2).

Figure 10. Triglyceride (TAG) level measurement in xenograft tumors. TAG levels were measured in tumors obtained from the xenograft models with indicated genotypes and treatment (n=4 to 5 tumor samples per group). *: P<0.05.

We acknowledge that the magnitude of TAG level changes observed in tumor samples was not as pronounced as that seen in cell line studies. This disparity could potentially be attributed to variations in the extent of cell cycle inhibition, with cell cycle inhibitor treatment in cell lines causing more dramatic suppression of cell proliferation than did 5-FU resistance in tumors. Nevertheless, we hope the reviewer will agree that these new results further substantiate the core conclusions drawn from our study.

Finally, it will be important for the authors to address the effects of the inhibition DGAT1/2 inhibition at the lipid levels. While the TAG formation would be suppressed, what happens to the rest of the lipidome? Where do the lipid production gets challenged or does the de novo synthesis gets shut down?

We appreciate the insightful comment from the reviewer. As mentioned earlier, during the revision of our manuscript, we repeated lipidomic analyses using LC-MS/MS. Our new data, presented in **Rebuttal Letter Figure 11A** (Figure 2i in the revised manuscript), confirmed that in vehicle-treated cells (those without nocodazole treatment), treatment with DGAT inhibitors led to anticipated outcomes: a decrease in the levels of TAG and TAG-O, and a corresponding increase in DAG. The impact of DGAT inhibition on other lipid species in vehicle-treated cells exhibited a dynamic nature, leading to increases in certain lipids while causing decreases in others.

Figure 11. Volcano plots in Caki-1 cells with comparison of iDGAT1/2 versus vehicle treatment without (A) or with nocodazole treatment (B).

Notably, the impact of DGAT inhibition on the lipidome changed significantly under nocodazole treatment conditions. In this context, the primary lipid species that exhibited a decrease were TAG and TAG-O, whereas the levels of most other lipid species, including various phospholipids, displayed increases (**Rebuttal Letter Figure 11B**; Fig. 2j in the revised manuscript).

These findings suggest that under nocodazole treatment conditions, DGAT inhibition does not seem to significantly affect de novo lipid synthesis but rather orchestrates a reshuffling of fatty acids, including PUFAs, from TAGs to various other phospholipid species (also see **Rebuttal Letter Figure 7b**).

References:

- Brandt, C., P.J. McFie, and S.J. Stone. 2016. Diacylglycerol acyltransferase-2 and monoacylglycerol acyltransferase-2 are ubiquitinated proteins that are degraded by the 26S proteasome. *Biochem J* 473:3621-3637.
- Choi, K., H. Kim, H. Kang, S.Y. Lee, S.J. Lee, S.H. Back, S.H. Lee, M.S. Kim, J.E. Lee, J.Y. Park, J. Kim, S. Kim, J.H. Song, Y. Choi, S. Lee, H.J. Lee, J.H. Kim, and S. Cho. 2014. Regulation of diacylglycerol acyltransferase 2 protein stability by gp78-associated endoplasmic-reticulum-associated degradation. *FEBS J* 281:3048-3060.
- Kagan, V.E., G. Mao, F. Qu, J.P. Angeli, S. Doll, C.S. Croix, H.H. Dar, B. Liu, V.A. Tyurin, V.B. Ritov, A.A. Kapralov, A.A. Amoscato, J. Jiang, T. Anthonymuthu, D. Mohammadyani, Q. Yang, B. Proneth, J. Klein-Seetharaman, S. Watkins, I. Bahar, J. Greenberger, R.K. Mallampalli, B.R. Stockwell, Y.Y. Tyurina, M. Conrad, and H. Bayir. 2017. Oxidized arachidonic and adrenic PEs navigate cells to ferroptosis. *Nature chemical biology* 13:81-90.
- Minami, J.K., D. Morrow, N.A. Bayley, E.G. Fernandez, J.J. Salinas, C. Tse, H. Zhu, B. Su, R. Plawat, A. Jones, A. Sammarco, L.M. Liao, T.G. Graeber, K.J. Williams, T.F. Cloughesy, S.J. Dixon, S.J. Bensinger, and D.A. Nathanson. 2023. CDKN2A deletion remodels lipid metabolism to prime glioblastoma for ferroptosis. *Cancer Cell* 41:1048-1060 e1049.
- Tarangelo, A., J. Rodencal, J.T. Kim, L. Magtanong, J.Z. Long, and S.J. Dixon. 2022. Nucleotide biosynthesis links glutathione metabolism to ferroptosis sensitivity. *Life Sci Alliance* 5:
- Wong, A., S. Chen, L.K. Yang, Y. Kanagasundaram, and K. Crasta. 2018. Lipid accumulation facilitates mitotic slippage-induced adaptation to anti-mitotic drug treatment. *Cell Death Discov* 4:109.
- Zou, Y., W.S. Henry, E.L. Ricq, E.T. Graham, V.V. Phadnis, P. Maretich, S. Paradkar, N. Boehnke, A.A. Deik, F. Reinhardt, J.K. Eaton, B. Ferguson, W. Wang, J. Fairman, H.R. Keys, V. Dancik, C.B. Clish, P.A. Clemons, P.T. Hammond, L.A. Boyer, R.A. Weinberg, and S.L. Schreiber. 2020. Plasticity of ether lipids promotes ferroptosis susceptibility and evasion. *Nature* 585:603-608.

REVIEWER COMMENTS

Reviewer #1 (Remarks to the Author):

The authors have sufficiently addressed my previous comments and concerns.

Reviewer #2 (Remarks to the Author):

The authors have adequately addressed most of the criticisms, however some points were not fully addressed which are listed below. Once these are appropriately addressed, I would recommend the manuscript for publication.

- Authors claim that the reshuffling of PUFAs from TAGs to phospholipids resensitizes arrested cells to ferroptosis. Due to the difficulty in generating fragments containing acyl-chains during MS/MS, it is not easy to assign the FA chains in PL and TAG by MS/MS. To strengthen their findings, it is recommended that they provide all or a portion of the MS/MS spectra of the PL and TAG with PUFA to facilitate the assignment of FA chain compositions.
- To enhance the transparency of their research, it is advisable that the authors make the lipidomics data, including the raw files generated by the mass spectrometer, available for sharing as a Source data file or supplementary data file.

Reviewer #3 (Remarks to the Author):

The authors addressed my comments and suggestions sufficiently. Thank you.

Detailed point-by-point response to the reviewers' comments:

Reviewer #1 (Remarks to the Author):

The authors have sufficiently addressed my previous comments and concerns.

We appreciate the support from this reviewer.

Reviewer #2 (Remarks to the Author):

The authors have adequately addressed most of the criticisms, however some points were not fully addressed which are listed below. Once these are appropriately addressed, I would recommend the manuscript for publication.

We thank the reviewer for the kind support. In the following reply, we have addressed the remaining comments from the reviewer.

- Authors claim that the reshuffling of PUFAs from TAGs to phospholipids resensitizes arrested cells to ferroptosis. Due to the difficulty in generating fragments containing acyl-chains during MS/MS, it is not easy to assign the FA chains in PL and TAG by MS/MS. To strengthen their findings, it is recommended that they provide all or a portion of the MS/MS spectra of the PL and TAG with PUFA to facilitate the assignment of FA chain compositions.

We appreciate the concern from the reviewer regarding the assignment of FA chains in PL and TAG through MS/MS due to the complexities of generating informative acyl-chain fragments. In response to this suggestion, we have included the MS/MS spectra data for 4 representative PLs (Figure 1) and 2 TAGs (Figure 2) in our revised manuscript. These spectra offer a comprehensive and detailed insight into the FA chain compositions. We think that the provided examples will significantly contribute to the transparency and understanding of our research findings.

Phospholipids

Figure 1. MS/MS spectra of phospholipids (PLs). A. phosphatidylcholine, PC (18:1_24:1). B. phosphatidylinositol, PI (20:4_20:4). C. phosphatidylethanolamine, PE (18:1_22:4) D. phosphatidylserine, PS (18:0_22:4).

Triacylglycerides

Figure 2. MS/MS spectra of triacylglycerides (TAGs). A. TG (16:0_20:2_22:2) and B. TG (18:1_18:1_20:4).

- To enhance the transparency of their research, it is advisable that the authors make the lipidomics data, including the raw files generated by the mass spectrometer, available for sharing as a Source data file or supplementary data file.

We have deposited the MS/MS raw data in the Zenodo database (<https://doi.org/10.5281/zenodo.10059819>) under the identifier 10.5281/zenodo.10059819. Once our manuscript is published, we will provide an update in our deposit by adding a link to our publication and making it freely available to the public.

We have stated under “data availability” section (see page 29), the mass spectrometry-based lipidomics raw data files and metadata have been deposited in the Zenodo database with the identifier 10.5281/zenodo.10059819. Additionally, we have included the analyzed lipidomic data as the Supplementary Table 2 in our revised manuscript.

Reviewer #3 (Remarks to the Author):

The authors addressed my comments and suggestions sufficiently. Thank you.

We appreciate the support from this reviewer.

REVIEWERS' COMMENTS

Reviewer #2 (Remarks to the Author):

The authors have satisfactorily addressed my criticisms and concerns. I would recommend the manuscript for publication.